# Early *Locus Coeruleus* noradrenergic axon loss drives olfactory dysfunction in Alzheimer's disease

Carolin Meyer[1], Theresa Niedermeier [1], Paul L. C. Feyen[1,2], Felix L. Strübing [1,2], Boris-Stephan Rauchmann [3,4], Katerina Karali[1], Johanna Gentz[1,2], Yannik E. Tillmann [1,5], Nicolas F. Landgraf [1,2,6], Svenja-Lotta Rumpf[1], Katharina Ochs[1,2], Karin Wind-Mark [4], Gloria Biechele [5,7], Jessica Wagner[1,8,9], Selim Guersel [4], Carolin I. Kurz [4], Meike Schweiger[4], Danilo Prtvar[1], Yuan Shi [1,2], Richard B. Banati [10], Guo-Jun Liu [10,11], Ryan J. Middleton[11], Gerda Mitteregger-Kretzschmar[2], Robert Perneczky [1,4,9,12,13], Thomas Koeglsperger[1,14], Jonas J. Neher [1,8,9], Sabina Tahirovic [1], Matthias Brendel [1,5,9,15], Jochen Herms [1,2,9,16] & Lars Paeger [1,2,9,16] ✉

Alzheimer's disease (AD) often begins with non-cognitive symptoms such as olfactory deficits, which can predict later cognitive decline, though the mechanisms remain unclear. Pathologically, the brainstem locus coeruleus (LC), the main source of the neurotransmitter noradrenalin (NA) modulating olfactory information processing is affected early. Here we show early and distinct loss of noradrenergic input to the olfactory bulb (OB) coinciding with impaired olfaction in an AD mouse model, before appearance of amyloid plaques. Mechanistically, OB microglia recognize and phagocytose LC axons. Reducing phagocytosis genetically preserves LC axons and olfaction. Prodromal AD patients display elevated TSPO-PET signals in the OB, similarly to *App^{NL-G-F}* mice. We further confirm early LC axon degeneration in post-mortem OBs in patients with early AD. Our findings reveal a mechanism linking early LC damage to hyposmia in AD, suggesting olfactory testing and neurocircuit imaging for early diagnosis and enable timely therapeutic intervention for Alzheimer's disease.

Alzheimer's disease is currently the most prevalent and devastating form of dementia, affecting millions of people worldwide[1]. Extracellular deposition of β-amyloid (Aβ), the formation of Aβ-plaques and the aggregation of microtubule-associated protein tau forming neurofibrillary tangles are the pathological hallmarks of AD[2]. While causal therapies are still not available, recent Aβ targeting antibody-therapies moderately improve cognitive decline in patients at early AD stages[3,4]. However, therapeutic success critically depends on the earliest possible diagnosis, warranting a detailed understanding of the mechanisms prior to the onset of the first cognitive symptoms.

The locus coeruleus (LC) noradrenergic (NA) system is affected particularly early in AD. It is the first site where aberrant tau hyperphosphorylation (pTau) is detected, putatively kickstarting the spread of tau throughout the CNS[5]. Consequently, past research has focused intensely on the effects of pTau on LC physiology, while the role of Aβ in LC dysfunction has attracted only scant attention. Forebrain NA is almost solely derived from the LC and, as a function of its widespread axonal projections, regulates a variety of physiological processes including arousal and attention, sleep-wake cycles, memory, energy homoeostasis, cerebral blood flow and sensory

**Fig. 1 | Early LC axon degeneration in the OB coincides with olfactory deficits.**
**a** LC-NA neurons project to the olfactory bulb (OB). The OB is composed of five different layers. The dashed box highlights the analysed region in the OB. Graphic modified from Claudi, F. (2020). Mouse Top Detailed. Zenodo. https://doi.org/10.5281/zenodo.3925997. **b** Immunostaining of LC axons (NET, magenta) in the OB of C57BL/6 J and $App^{NL-G-F}$ mice at 1, 2, 3 and 6 months of age. Scale bar: 50 μm. **c** Relative NET fibre density. (C57BL/6 J at 1,2,3,6 months: $n = 4,5,6,7$ and $App^{NL-G-F}$ at 1,2,3,6 months: $n = 5,4,6,8$). **d** Absolute NET fibre density in different OB layers at 3 months of age. (Per layer: $n = 5$ C57BL/6 J vs. $n = 6$ $App^{NL-G-F}$). **e** Immunostaining of microglia (Iba1, green) and Aβ-plaques (Aβ, red). Scale bar: 50 μm. **f** Quantification of relative microglia density and (C57BL/6 J at 1,2,3,6 months: $n = 5,5,6,7$ and $App^{NL-G-F}$ at 1,2,3,6 months: $n = 5,4,6,8$) (**g**) total Aβ-plaque load in $App^{NL-G-F}$ mice. (1 vs. 2 months $n = 4$ vs. 4; 2 vs. 3 months $n = 4$ vs. 6; 3 vs .6 months $n = 6$ vs. 8). **h** Representative confocal images of TH-positive LC neurons (magenta) and

Aβ-plaques (red). Scale bar: 50 μm. **i** Relative LC neuron number in 12-month-old C57BL/6 J and $App^{NL-G-F}$ mice. ($n = 3$ vs. 3). **j** Olfactory tests used in study. **k** Time to find food in the buried food task at 1,3, and 6 months of age. (C57BL/6 J at 1,3,6 months: $n = 9,14,14$ and $App^{NL-G-F}$ at 1,3,6 months: $n = 10,18,24$.) **l** Exemplary traces of distance versus time animals spend interacting with a low (1:1000) and a high (1:1) vanilla odour concentration at 3 months of age. **m** Time mice spend in the investigation zone (<2 cm to cotton tip). (low/high vanilla for C57BL/6 J: $n = 10/9$ and low/high vanilla for $App^{NL-G-F}$: $n = 11/10$). **n** Number of entries in investigation zone (C57BL/6 J vs. $App^{NL-G-F}$: $n = 9$ vs. 9). Data expressed as mean ± s.e.m.; ns, not significant; *$p < 0.05$, **$p < 0.01$, ****$p < 0.0001$; unpaired, two-tailed t-test; 4 slices per animal in (**c, d, f, i** (8 slices) **and k**); One-way ANOVA with Sidak's post-hoc test, 4 slices per animal in (**g**), Two-way ANOVA with Tukey's post-hoc test in (**m**); Two-way ANOVA with Sidak's post-hoc test in (**n**); Statistics shown in Supplementary Data 1. Source data are provided as a Source Data file.

processing, all of which are impaired in the progression of AD, though with differences in temporal progression[6]. Symptomatically, early olfactory dysfunction frequently marks the early onset of AD, with prospective patients remaining cognitively normal and otherwise healthy[7,8]. Although decreased olfactory sensitivity is apparent in ~85% of AD cases, the underlying mechanisms remain a conundrum[9,10]. Here, we ventured for a multifaceted approach to study the neural correlate of olfactory dysfunction in a mouse model of amyloidosis using a plethora of steady-state systems neuroscience techniques, both ex vivo and in vivo and studied human post-mortem brain tissue to validate our mechanistic findings. In this work, we show that early LC axonal degeneration occurs exclusively in the OB of $App^{NL-G-F}$ mice. This is dependent on OB microglia, recognising externalised phosphatidylserine on LC axons and results in olfactory deficits in these animals. Early gliosis can be detected in human prodromal AD patients, and LC axon density decreased in post-mortem tissue from early AD patients. Collectively, we reveal a mechanistic link between early olfactory deficits and LC vulnerability in AD. Our work may help to facilitate early diagnosis and intervention.

## Results

### Early LC axon loss exclusive to the OB

LC axon loss has been reported at late disease stages in the $App^{NL-G-F}$ mouse model[11]. By systematic comparison of multiple brain areas, we set out to analyse if LC axon loss might already be detected earlier in these animals (Fig. 1a). Surprisingly, we discovered an early LC axon degeneration exclusive to the OB starting between 1 and 2 months in $App^{NL-G-F}$ mice (Fig. 1a–d). While in 1-month-old animals, the LC axon density was unaltered compared to WT animals, we observed a 14% fibre loss at 2 months of age. This loss further progressed to 27% at 3 months, and 33% at 6 months. Notably, LC axons started to degenerate in other regions such as the hippocampus, piriform cortex and medial prefrontal cortex between 6 and 12 months at the earliest (Supplementary Fig. 1a, b). Since the LC-NA system is not the only subcortical modulatory system known to innervating the OB and to be affected early in AD, we also assessed cholinergic and serotonergic projections. Importantly, we did neither detect a decreased density of choline-acetyl-transferase (ChAT[+]) nor of serotonergic transporter (SERT[+]) neurites at the age of 3 months (Supplementary Fig. 1e–g). We thus concluded that the loss of axons in the OB is specific to the LC-NA

system at this age. Similar to the cortex, the OB is composed of different layers, which are disparately innervated by the LC-NA system. We thus analysed layer-specific axon loss and identified the most densely innervated region, the internal plexiform layer, to be the site of most prominent axon loss, followed by the external plexiform layer (Fig. 1d). OB microglia increased between 2 and 3 months of age without significant Aβ plaque deposition (Fig. 1f, g). We excluded NA cell loss in the LC to underlie axonal demise, as we did not observe differences in LC neuron number in $App^{NL-G-F}$ mice when compared to WT animals at 12 months (Fig. 1h, i). We next asked whether early deposition of extracellular Aβ correlates with LC axonal damage. Intriguingly, we found LC fibre loss to be independent of the amount of extracellular Aβ (Supplementary Fig. 2a).

## LC axon loss drives hyposmia

Early sensory manifestations such as hyposmia have been well described in prodromal AD (pAD), as have the contributions of NA to olfaction[12]. Thus, we set out to analyse whether LC axon loss results in impaired olfaction. We employed the buried food test, a well-established olfactory task to measure the ability of an animal to detect volatile odours[13] (Fig. 1j). Food-deprived WT animals rapidly started exploring the arena and usually uncovered the hidden food pellet within ~40 s. In contrast, 3-month-old $App^{NL-G-F}$ mice needed 60% more time to find the buried food pellet. The same phenotype was reproduced in 6-month-old animals (Fig. 1k). We did not observe any differences when testing animals at 1 month of age, which is consistent with the lack of LC axon degeneration at that time point (Fig. 1b, c, k). To rule out task-specific confounders, we aimed to recapitulate our findings in a second olfactory task. To this end, we subjected 3-month-old WT and $App^{NL-G-F}$ mice to an odour sensitivity test (Fig. 1l–n). We exposed the animals to ascending concentrations of vanilla, a pleasant odour, and measured the time the animals spent interacting with the odour delivery stick (Fig. 1m, n). WT animals were readily attracted by a low odour concentration (dilution 1:1000) and repeatedly interacted with the odour stick, while $App^{NL-G-F}$ mice visited the interaction zone considerably later and less often. The same behaviour was observed when testing a high vanilla concentration (dilution 1:1; Fig. 1m, n). Collectively, these data reveal a consistent olfactory phenotype in $App^{NL-G-F}$ mice, starting at 3 months of age, which is hitherto the earliest behavioural manifestation described in this mouse model.

## Impaired NA release links to hyposmia

Neurocircuit-homoeostasis is able to partially balance molecular and structural changes or loss in case of neuropathological insults[14]. We thus aimed to understand whether LC axon loss translates into decreased NA release in the OB. In order to investigate potential changes in the concentration of NA in the OB of $App^{NL-G-F}$ animals, we performed an NA ELISA. Interestingly, we did not observe a significantly different concentration of baseline NA in these animals compared to WT mice (Supplementary Fig. 3a). We thus hypothesised that a change in LC-NA would be more pronounced in stimulus-related NA release. We transduced the OB of 2-month-old WT and $App^{NL-G-F}$ animals with the NA sensitive biosensor $GRAB_{NE}$ (G-protein-coupled receptor-activation-based sensor for noradrenaline) or its mutant control ($GRAB_{NE(mutant\ ctrl)}$ and implanted a chronic cranial window over the olfactory bulb (Fig. 2a)[15]. At 3 months of age, we performed in vivo acousto-optical 2-photon (AO-2P) microscopy in awake animals paired with olfactory stimulation by 10 s long vanilla puffs (Fig. 2b–g). WT animals reliably and repeatedly responded to the odour delivery with a strong and long-lasting increase of fluorescence, unlike animals injected with the mutant control sensor, as measured over the entire field of views (FOV; Fig. 2f–h). As a control, neither a blank air puff nor NA measurements in the cortex coupled to odour delivery elicited coherent changes in fluorescence (Fig. 2d and Supplementary Fig. 3b). To account for potential differential dynamics (increase vs. decrease) of NA, we binned each of the three FOVs into 36 regions of interest

(ROIs). Plotting each individual ROI revealed a striking increase in NA release upon odour-stimulation. In WT animals, 75% ROIs showed an increase in NA, while only 5% showed a decrease. This relationship was altered in $App^{NL-G-F}$ mice (Fig. 2i). We furthermore sought to investigate if impaired NA release is independent of the given odour. In addition to Vanilla, we thus stimulated with Lemon and assessed odour-evoked NA release in a separate cohort using the chemically defined odorants Isoamylacetate (Banana) and (S)-(+)-Carvon (Caraway). Interestingly, the decrease in NA release in $App^{NL-G-F}$ mice compared to WT animals was apparent in all odours tested (Fig. 2j, l) and absent in a blank control where only mineral oil was used (Fig. 2k). Immunohistochemical validation revealed a solid transduction of the tissue in the OB of all animals and NA fibre loss in $App^{NL-G-F}$ mice (Fig. 2m, n). To exclude the possibility of dysfunctional mitral cells, the first-order projection neurons of the OB, driving impaired olfaction, we performed perforated patch-clamp recordings of mitral cells in acute OB slices. In line with previous studies, we found mitral cells to be spontaneously active, but we did not detect alterations of intrinsic properties between genotypes at 6 months of age, at which hyposmia is well manifested in these animals (Supplementary Fig. 4a–f)[16]. Recent sophisticated work analysed the downstream effect of NA release on mitral cells by in vivo electrophysiology combined with optogenetics[17]. This revealed a complex pattern of each a third of neurons being excited, inhibited or unresponsive to NA. In line, when we applied exogenous NA (30 μm) and recorded the change in action potential (AP) frequency of mitral cells, we discovered differential effects on mitral cell membrane potential in both WT and $App^{NL-G-F}$ mice (Supplementary Fig. 4g–i). Of note, a slight trend toward a decreased responsiveness of mitral cells in $App^{NL-G-F}$ mice could be detected when comparing F-I relationships with and without the presence of administered NA, which might be indicative for a homoeostatic downregulation of adrenergic receptors upon decreased NA release (Supplementary Fig. 4j). The structure-to-function relationship of the LC-NA system and olfaction led us to further probe whether persistent activation of remaining LC axons by chemogenetics would be sufficient to reinstate olfaction (Supplementary Fig. 5a–c). We bilaterally injected an AAV transducing LC neurons of $App^{NL-G-F}$ x $Dbh$-$Cre$ animals with an excitatory ligand-gated G-protein-coupled receptor (h3MDGs, designer-receptor exclusively activated by designed drugs, DREADD). In patch-clamp recordings, we confirmed that the application of Clozapine-N-Oxide (CNO) readily activates LC neurons (Supplementary Fig. 5a, b), however, neither acute nor prolonged (repetitive delivery of clozapine instead of CNO) systemic administration to activate excitatory DREADDs in vivo was sufficient to accelerate the time to find the buried food pellet in $App^{NL-G-F}$ x $Dbh$-$Cre$ mice (Supplementary Fig. 5a–f). This strongly suggests a structure-to-function relationship of LC axons in the OB in the context of olfaction.

## OB microglia clear LC axons

Microglia have been attracting considerable attention in the pathogenesis of AD[18]. Their remarkable heterogeneity has been revealed recently, highlighting the complex nature of microglia and their influence on brain functions[19]. Since early LC axon loss coincides with an increased number of microglia, we set out to investigate whether microglia could account for LC axon loss. Thus, we performed bulk RNA sequencing (RNA-seq) of microglia isolated from OBs of WT and $App^{NL-G-F}$ mice at the age of 2 months, the very onset of LC axon loss (Fig. 3a). In line with our immunohistological data, we observed an increased number of microglia cells isolated from bulbi of $App^{NL-G-F}$ animals (Fig. 3b). After appropriate quality control (Supplementary Fig. 6), we performed differential expression testing using negative binomial models while controlling for sex. This revealed that 2.344 genes (of a total of 17.840) were differentially expressed, with a slight majority of them (1.283) being upregulated in $App^{NL-G-F}$ animals (Fig. 3c

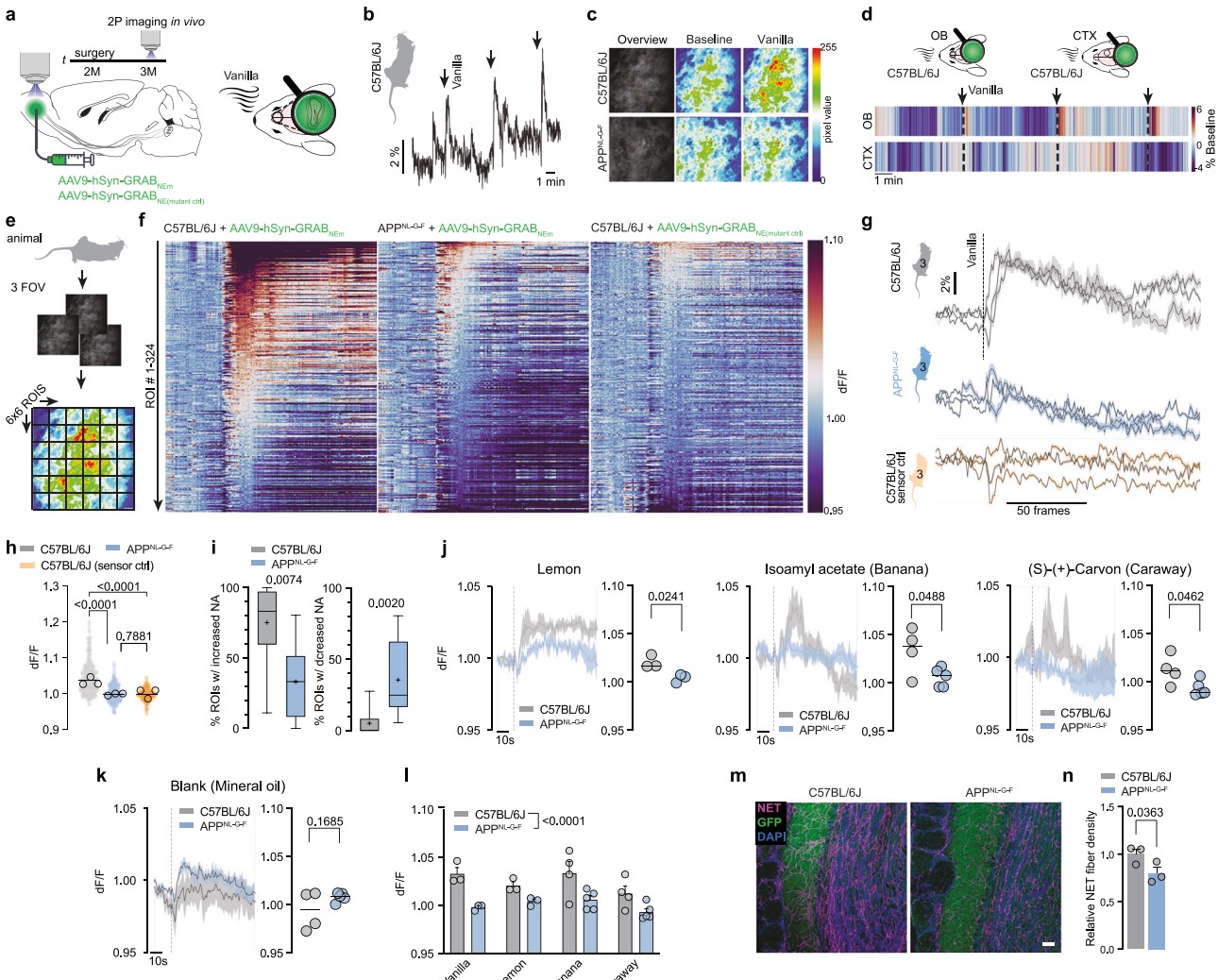

**Fig. 2 | Decreased odour-stimulated noradrenaline release in the OB of $App^{NL-G-F}$ mice in vivo. a** Experimental setup of noradrenaline (NA) level measurements in vivo. Graphic modified from Carpaneto, A. (2020). Microscope Objective. Zenodo. https://doi.org/10.5281/zenodo.3926119, Petrucco, L. (2020). Mouse head schema. Zenodo. https://doi.org/10.5281/zenodo.3925903. **b** NA response of a C57BL/6 J mouse to three consecutive vanilla air puffs. Graphic modified from Claudi, F. (2020). Mouse Top Detailed. Zenodo. https://doi.org/10.5281/zenodo.3925997. **c** Exemplary images and heat map of baseline and odour-induced NA release in the OB, taken from a C57BL/6 J and $App^{NL-G-F}$ animal. **d** NA release measured in the OB and cortex (CTX) of a C57BL/6 J mouse following three consecutive vanilla air puffs in comparison to the same stimuli in the OB. Graphic modified from Petrucco, L. (2020). Mouse head schema. Zenodo. https://doi.org/10.5281/zenodo.3925903. **e** Illustration of the analysis of 2 P in vivo imaging data. Graphic modified from Claudi, F. (2020). Mouse Top Detailed. Zenodo. https://doi.org/10.5281/zenodo.3925997. **f** Heat maps of NA response to one vanilla air puff comparing C57BL/6 J mice vs. $App^{NL-G-F}$ mice and C57BL/6 J expressing the mutant NA sensor control. **g** Grand average per animal from all 324 ROIs depicted in f. Graphic

modified from Claudi, F. (2020). Mouse Top Detailed. Zenodo. https://doi.org/10.5281/zenodo.3925997. **h** Distribution of all rel. changes in fluorescence for the three groups ($n = 3$ per group). **i** Fraction of ROIs responding with an increase or decrease in fluorescence ($n = 3$ C57BL/6 J vs. $n = 3$ $App^{NL-G-F}$). **j** NA release during stimulation with further odours. (Lemon: $n = 3$ vs. 3, Banana: $n = 4$ vs. 5, Caraway: $n = 4$ vs. 5). **k** NA imaging with blank (mineral oil) stimulation. ($n = 4$ vs. 5). **l** Overall decrease of NA release upon odour-stimulation across all tested odours. **m** Representative confocal images of virus expression (GPF, green) and LC axon density (NET, magenta) in the OB. Scale bar: 50 μm. **n** Relative NET fibre density at 3 months of age ($n = 3$ vs. 3); Data expressed as mean ± s.e.m.; *$p < 0.05$, **$p < 0.01$; Kruskal-Wallis test with Dunn's multiple comparison test in (**h**); two-tailed Mann-Whitney test in (**i**); Unpaired, two-tailed t-test in (**j**, **k**, **n**). Mixed effects analysis in (**l**), genotype ($F(1,22) = 27,12$), Box plots show: 50th percentile (median value, line; mean value, +), 25th to 75th percentiles of dataset (box), 5th and 95th percentile (Whiskers)). Statistics shown in Supplementary Data 1. Source data are provided as a Source Data file.

and Supplementary Data 1). Previous work has demonstrated a so-called "disease-associated" microglia response (DAM) in AD mouse models and humans alike[20,21]. To test whether this phenotype was visible in our data, we directly compared our microglia OB RNA-seq data to a publicly available cortical microglia RNA-seq dataset taken from 8-month-old $App^{NL-G-F}$ mice[22]. Linear regression of log-fold changes in fact revealed a significant negative relationship ($R = -0.44$, $p < 2e-16$) between young OBs and aged cortex. While no significant relationship was found when filtering for known DAM- or homoeostatic microglial genes, the vast majority of the ~3100 genes that both

datasets had in common are unannotated (Fig. 3d). Nevertheless, even though this relationship was driven by those genes with unknown functions, our results still suggest that the biological state of microglia in young OBs is distinct from older cortices (Fig. 3d). A crucial function of microglia is the removal of debris or apoptotic cells from the parenchyma as well as synaptic remodelling[23]. Interestingly, gene ontology (GO) term analysis revealed the 20 most enriched terms relate to neuronal function and synaptic or neuronal plasticity. We compared all identified transcripts annotated to the GO term "Phagocytosis". We identified 121 transcripts, of which surprisingly only 2 were

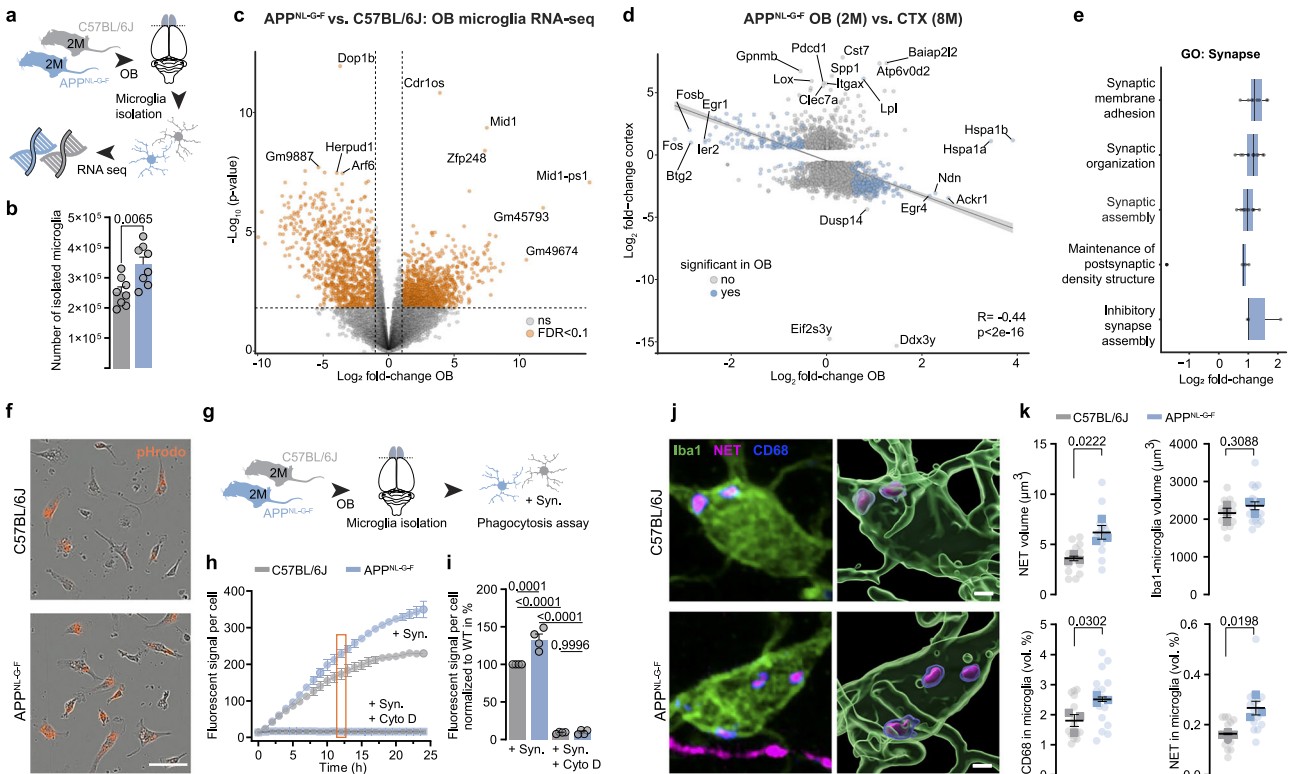

**Fig. 3 | Increased *App^NL-G-F* microglia phagocytosis of LC axons in the OB.**
**a** Experimental setup of RNA sequencing from OB microglia of 2-month-old animals. Graphic modified from Claudi, F. (2020). Mouse Top Detailed. Zenodo. https://doi.org/10.5281/zenodo.3925997, Thompson, E. (2020). Mouse Brain Above. Zenodo. https://doi.org/10.5281/zenodo.3925971 and Chilton, J. (2020). Microglia resting. Zenodo. https://doi.org/10.5281/zenodo.3926033. **b** Number of isolated microglia. (*n* = 8 C57BL/6 J vs. *n* = 8 *App^NL-G-F*). **c** Volcano plot visualising differentially expressed microglia genes (orange). **d** Volcano plot comparing microglia genes from the OB of 2-months-old *App^NL-G-F* mice to the cortex of 8-months-old *App^NL-G-F* mice (Sobue et al., 2021). **e** Gene ontology (GO) enrichment analysis of genes involved in synapses. **f** Microglia cell pictures taken with the Incucyte live-cell analysis system after 12 h incubation with synaptosomes (pHrodo, orange). Scale bar: 50 μm. **g** Experimental design for phagocytosis assay. Graphic modified from Claudi, F. (2020). Mouse Top Detailed. Zenodo. https://doi.org/10.5281/zenodo.3925997, Thompson, E. (2020). Mouse Brain Above. Zenodo. https://doi.org/10.5281/zenodo.3925971 and Chilton, J. (2020). Microglia resting. Zenodo.

https://doi.org/10.5281/zenodo.3926033. **h** pHrodo fluorescent signal per cell over 24 h comparing phagocytotic activity of C57BL/6 J and *App^NL-G-F* microglia. **i** Fluorescent signal per cell normalised to C57BL/6 J at the time point 12 h. (*n* = 4 C57BL/6 J vs. *n* = 4 *App^NL-G-F*, each 3 technical replicates). **j** Immunostaining and 3D reconstruction of microglia (Iba1, green), lysosomes (CD68, blue) and LC axons (NET, magenta). Scale bar: 2 μm. **k** Analysis of NET volume, Iba1 volume and CD68 volume. *App^NL-G-F* microglia contain more NET⁺ signal than C57BL/6 J microglia (*n* = 3 C57BL/6 J vs. *n* = 3 *App^NL-G-F*, each 5 technical replicates); Data expressed as mean ± s.e.m.; ns, not significant; *$p < 0.05$, **$p < 0.01$, ***$p < 0.001$, ****$p < 0.0001$; Unpaired, two-tailed t-test in (**b**, **k**); One-way ANOVA with Tukey's post-hoc test in (**k**); Fig. 3d and e do not contain probability tests. Figure 3c depicts the results of a differential gene expression analysis from a quasi-likelihood negative binomial generalised log-linear model fitted to count data, and was corrected for multiple comparisons using the Benjamini-Hochberg FDR. Statistics shown in Supplementary Data 1. Source data are provided as a Source Data file.

differentially expressed in our data set (Supplementary Fig. 7a). However, when analysing gene modules related to the GO-term "synapse", we observed an overarching upregulation of 73 genes, suggesting an increased plastic environment, potentially indicating increased synaptic pruning (Fig. 3e). Indeed, several genes included in this cluster are suggested to also play directly or indirectly a role in phagocytosis (*Chrna7, Lrrtm4, Bln2, Epha4*)[24–28]. We thus hypothesised that microglia phagocytosis might be responsible for the selective clearance of LC axons in the olfactory bulb. Thus, we conducted an automated phagocytosis assay from primary OB microglia of WT and *App^NL-G-F* mice, aged 2 months. Microglia were incubated with pHrodo-labelled synaptosomes to measure their phagocytic uptake over the course of 24 h (Fig. 3f–i). Our data revealed an increased efficiency of *App^NL-G-F* microglia to phagocytose fluorescently labelled synaptosomes, with OB microglia of *App^NL-G-F* mice showing a 33% higher phagocytic capacity already after 12 h. As expected, Cytochalasin-D application completely abolished phagocytosis in both genotypes (Fig. 3h, i). Based on their increased phagocytic activity, we hypothesised that microglia might indeed be phagocytosing LC axons in OBs from *App^NL-G-F* mice. To test this directly, we performed high-resolution

imaging of NET fibres together with microglia and the lysosomal marker CD68 and subsequently performed 3D-reconstructions of these images (Fig. 3j). We found a higher volume of NET⁺ immuno-signal in single microglia cells from *App^NL-G-F* mice compared to WT animals, as well as increased volumes of lysosomal CD68 (Fig. 3k), corroborating the increase in phagocytic activity observed in vitro. Notably, we did not see significant differences in the cellular volumes of single microglia between groups. Collectively, our data show no overt disease-associated activation of microglia, but a strikingly increased phagocytic activity compared to WT animals of the same age. Consequently, we hypothesised that an inhibition of phagocytosis could prevent the loss of NA axons in the OB. Translocator protein 18 kDa (TSPO) has recently been identified as a key protein in fuelling synaptic pruning and microglial phagocytosis[29–31]. TSPO-KO decreases ATP production associated mitochondrial functions and the innate immune processes of microglia and ultimately reduced phagocytosis[32,33]. Based on these findings, we sought to investigate if TSPO elimination would be sufficient to halt or decelerate the loss of LC axons. To this end, we bred mice with a global knockout of TSPO[33] to *App^NL-G-F*. We again harvested OBs from these animals at 2–6 months

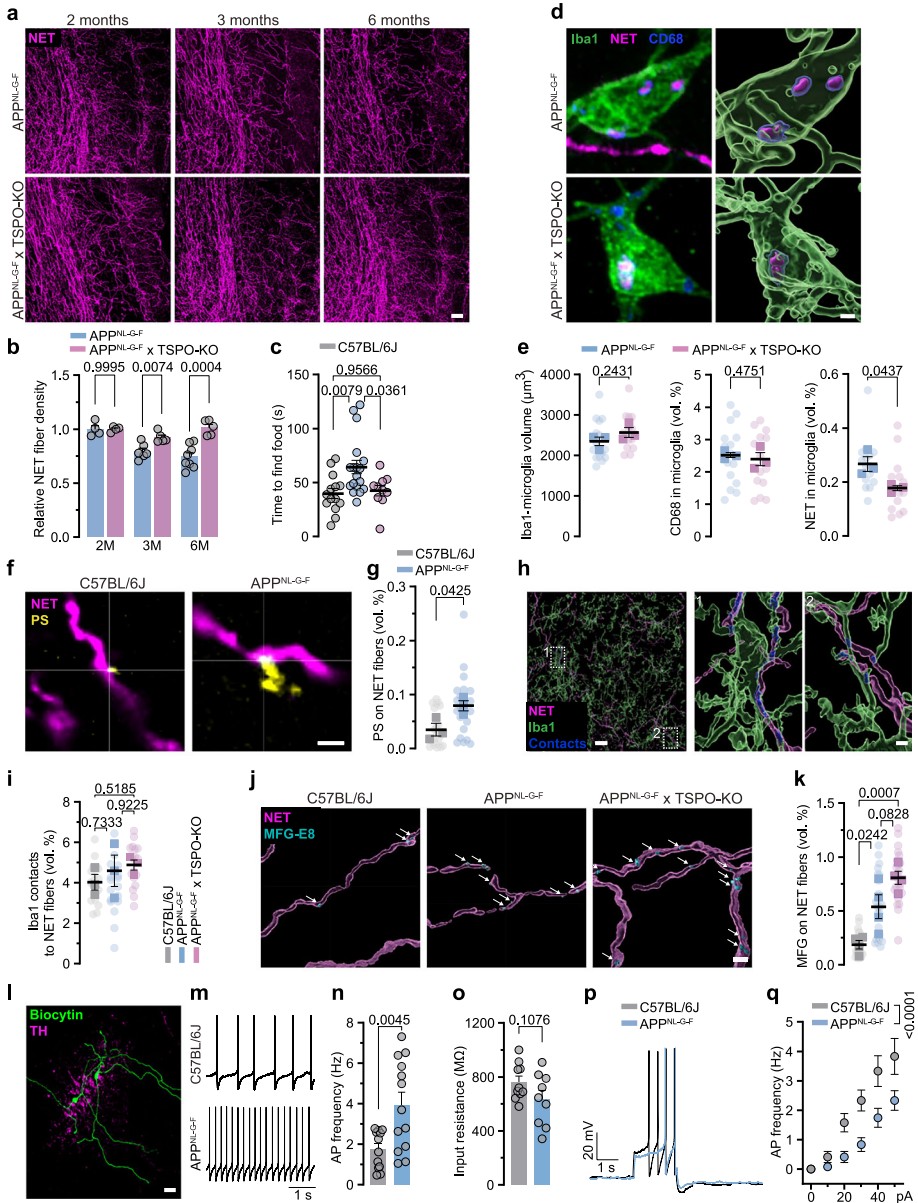

**Fig. 4 | Reduced phagocytosis rescues axons and hyposmia, caused by PS-MFG-E8 axon decoration. a** Immunostaining of LC axons (NET, magenta) in the OB of *App*^NL-G-F mice and *App*^NL-G-F x TSPO-KO mice at 2, 3 and 6 months of age. Scale bar: 50 μm. **b** Relative NET fibre density. (C57BL/6 J at 2,3,6 months: *n* = 4,6,8 and *App*^NL-G-F at 2,3,6 months: *n* = 4,5,5). **c** Buried food test comparing the time to find a food pellet is rescued in *App*^NL-G-F x TSPO-KO mice at 3 months of age. (C57BL/6 J: *n* = 14, *App*^NL-G-F: *n* = 18, *App*^NL-G-F x TSPO-KO: *n* = 10). **d** Immunostaining and 3D reconstruction of microglia (Iba1, green), lysosomes (CD68, blue) and LC axons (NET, magenta). Scale bar: 2 μm. **e** Analysis of NET volume, Iba1 volume and CD68 volume. *App*^NL-G-F x TSPO-KO microglia contain less NET^+ signal than *App*^NL-G-F microglia. (*n* = 3 C57BL/6 J vs. *n* = 3 *App*^NL-G-F, each 5 pictures). **f** Immunostaining visualising LC axons (NET, magenta) tagged with phosphatidylserine (PS, yellow). Scale bar: 2 μm. **g** Percental volume of PS colocalised with NET fibres. (*n* = 3 C57BL/6 J vs. *n* = 3 *App*^NL-G-F, each 6 pictures). **h** Contact points (blue) between microglia (Iba1, green) and LC axons (NET, magenta). Scale bar: 20 μm, zoom in: 2 μm. **i** Quantification of Iba1-LC axon contact points. (C57BL/6 J: *n* = 3, *App*^NL-G-F: *n* = 3,

*App*^NL-G-F x TSPO-KO: *n* = 3, each 6 pictures). **j** 3D reconstruction of MFG-E8 adaptor protein (MFG-E8, cyan) colocalised to LC axons (NET, magenta). Scale bar: 2 μm. **k** Analysis of MFG-E8 volume colocalised to LC axons. (C57BL/6 J: *n* = 4, *App*^NL-G-F: *n* = 4, *App*^NL-G-F x TSPO-KO: *n* = 4, each 6 pictures). **l** Confocal image showing two biocytin-filled neurons (green) of the LC (TH, magenta). Scale bar: 20 μm. **m** Representative traces of spontaneous action potential firing. **n** Quantification of action potential frequency. (C57BL/6 J: *n* = 8 (12 cells), *App*^NL-G-F: *n* = 10 (13 cells)). **o** Input resistance. (C57BL/6 J: *n* = 8 (10 cells), *App*^NL-G-F: *n* = 9 (9 cells)). **p** Representative traces of evoked action potentials (at 50 pA current injections). **q** Current-frequency curve showing LC neurons from *App*^NL-G-F mice to be less excitable (C57BL/6 J: *n* = 8 (12 cells), *App*^NL-G-F: *n* = 9 (11 cells)); Data expressed as mean ± s.e.m.; ns, not significant; *$p < 0.05$, **$p < 0.01$, ***$p < 0.001$, ****$p < 0.0001$; Unpaired, two-tailed t-test in (**b**, **e**, **g**, **n**, **o**); One-way ANOVA with Tukey's host-hoc test in (**c**, **i**, **k**); Two-way ANOVA with Sidak's post hoc test in (**q**); Statistics shown in Supplementary Data 1. Source data are provided as a Source Data file.

of age and stained for NET^+ LC axons. Indeed, the lack of TSPO in *App*^NL-G-F mice abrogated the loss of NA axons in these animals up to an age of 6 months (Fig. 4a, b). This correlated with a decreased uptake of NET^+ axons in microglia of TSPO-KO x *App*^NL-G-F mice (Fig. 4d, e). We then exposed the TSPO-KO x *App*^NL-G-F animals to the buried food task. Importantly, the preservation of LC axons in the OB resulted in a

retained ability to find the buried food pellet indistinguishable from WT animals (Fig. 4c).

## PS labels LC axons for phagocytosis

A plethora of "find-me"- and "eat-me"-signals attracting microglia to their phagocytic targets have been revealed within the last years[34]. The

complement cascade has emerged as one key player of synaptic removal in AD[35]. We thus aimed to analyse whether LC axons from $App^{NL-G-F}$ mice would be decorated by Complement component 1q (C1q) as a possible underlying cause of axonal clearance. As expected, staining for C1q resulted in a dense punctate pattern. However, we did not observe any significant changes of C1q colocalization to NET+ axons in the OBs of $App^{NL-G-F}$ mice compared to WT mice (Supplementary Fig. 8a, b). In both healthy and diseased brains, the highly coordinated local externalisation of phosphatidylserine (PS) leads to the targeted engulfment of neuronal material by microglia and has similarly been described to contribute to synapse loss in AD mouse models[36,37]. A variety of microglial receptors are known to recognise exposed PS, such as triggering receptor expressed in myeloid cells 2 (TREM2) and milk fat globule-EGF factor 8 protein (MFG-E8), which in turn bind to microglial vitronectin receptors (the $\alpha_v\beta3/5$ integrins), both of which play major roles in the aetiology of AD[36,38,39]. While PS recognised by TREM2 was shown to contribute to synapse loss in $App^{NL-G-F}$ mice, PS and MFG-E8 are important physiological mediators of microglia-dependent synaptic pruning during adult neurogenesis in the OB of mice[36]. Considering the increase of mRNAs associated with synaptic plasticity (Fig. 3e), we hypothesised that increased PS externalisation might be the underlying cause of LC axon phagocytosis by microglia. To test this, we performed in vivo PS labelling by injecting PSVue550 in the OBs of WT and $App^{NL-G-F}$ mice at the age of 5 months. Importantly, as shown previously and in line with its physiological function, we could visualise externalised PS in the OB, both in WT and $App^{NL-G-F}$ mice. In order to assess whether PS externalisation can be detected on NET+ axons, we conducted a colocalization analysis using 3D reconstruction. When adjusting for the fibre density, we found an elevated colocalization of PS on NET+ axons in $App^{NL-G-F}$ mice (Fig. 4f, g). Intriguingly, flipped PS was often accompanied by Iba1+ microglia directly contacting LC axons. However, when analysing the contact points between microglia and LC axons, no statistical difference in colocalised volume was found between the genotypes, although a tendency toward an elevation could be observed (Fig. 4h, i). Further investigating the possible link, we could show that PS is capped with MFG-E8, serving as the adaptor protein between PS and the microglial integrin receptor (Supplementary Fig. 9a). Using 3D reconstruction, we found more MFG-E8 colocalised to LC axons of $App^{NL-G-F}$ mice than on LC axons from WT animals (Fig. 4j, k). Given the TSPO-KO-mediated rescue of LC axons and olfaction, we hypothesised that MFG-E8 decoration should similarly be increased in $App^{NL-G-F}$ x TSPO-KO mice. We stained OB tissue from these animals for LC axons and MFG-E8 and again reconstructed both signals. Intriguingly, MFG-E8 decoration of LC axons was clearly increased compared to WT animals and even showed a trend towards an increase compared to $App^{NL-G-F}$ mice (Fig. 4j, k). Overall, we conclude that local PS externalisation in conjunction with MFG-E8 decoration constitutes a major "eat-me" signal for microglia interaction with LC axons and subsequent phagocytosis. We finally ventured to elucidate, mechanistically as to why PS is externalised on LC axons. In neurons, the protein TMEM16F constitutes a $Ca^{2+}$-dependent scramblase responsible for PS externalisation. Earlier work has put much emphasis on the firing properties of LC neurons and the $Ca^{2+}$-dependence of their intrinsic pacemaker, especially in the context of neurodegeneration[40]. During the pacemaking activity of LC neurons, each action potential (AP) is accompanied by a $Ca^{2+}$-driven suprathreshold oscillation, which leads to the activation of voltage-gated sodium channels underlying the super-threshold AP. We thus hypothesised that increased firing in LC neurons may underlie $Ca^{2+}$-triggered scramblase to flip PS to the outside of the plasma membrane. We performed perforated patch-clamp recordings of LC neurons from WT and $App^{NL-G-F}$ mice at the age of 6 months (Fig. 4l–q). Indeed, we found an overall increase in spontaneous AP frequency in acute brain slices from $App^{NL-G-F}$ (Fig. 4m, n). We did not observe a change in input resistance during hyperpolarisation but a slightly decreased intrinsic

excitability in response to depolarising stimuli, likely reflecting an increased activation of $Ca^{2+}$-dependent potassium channels (Fig. 4o–q). We thus conclude that spontaneous hyperactivity in LC neurons and consequently elevated $Ca^{2+}$-signalling instigates $Ca^{2+}$-dependent scramblase/flippase, leading to the externalisation of PS and a microglia-mediated removal of hyperactive LC originating axons. In summary, we clearly pinpoint microglial phagocytosis of NA axons in the OB as the underlying cause of the progressive early axon loss in $App^{NL-G-F}$ mice.

## LC-$App^{NL-G-F}$ expression induces hyposmia

In $App^{NL-G-F}$ mice, every $App$-expressing cell harbours three mutations, limiting the conclusion about the relative effect of LC axon loss[41]. Thus, we asked whether $App^{NL-G-F}$ expression restricted to the LC would be sufficient to recapitulate the neuroanatomical and behavioural findings. We engineered a custom-built Cre-dependent AAV to specifically transduce LC neurons of Dbh-Cre mice with the human $App^{NL-G-F}$ ($Dbh$-$hApp^{NL-G-F}$) or a control virus, leading to the expression of a fluorophore only ($Dbh$-$EYPF$; Fig. 5a). Three-months post-injection, we performed a buried food test. Of note, $Dbh$-$hApp^{NL-G-F}$ mice needed more time to find the buried food compared to the control injected $Dbh$-$EYPF$ mice (Fig. 5d, e). Immunohistochemical validation revealed an LC axon degeneration of 15% in the OB of $Dbh$-$hApp^{NL-G-F}$ mice compared to $Dbh$-$EYPF$ mice (Fig. 5b, c), without LC neuron loss (Supplementary Fig. 10a, b). We thus asked next whether again microglia in the OB would phagocytose LC axons and performed the same set of immunohistological staining to assess NET protein within CD68+ lysosomes of microglia. Indeed, we observed an increase in the volume of NET+ signal inside the lysosomes of microglia (Fig. 5f, g). Collectively, our approach to induce $Dbh$-$hApp^{NL-G-F}$ expression specifically in LC neurons illustrates that this is sufficient to recapitulate both early behavioural and neuropathological phenotypes observed in the $App^{NL-G-F}$ mouse line.

## LC axon loss and hyposmia in human pAD

Early impairment of the LC-NA system in humans has recently been in the spotlight of several multimodal imaging studies[42]. While at the level of the brainstem, LC volume decreases over time and levels of LC integrity predict cognitive outcome in elderly subjects, it is not yet clear whether axon loss also precedes late-phase occurring cell loss in the LC of humans[43]. Interestingly, both hyposmia and LC integrity are predictors of cognitive decline in humans[7,8]. We thus ventured to decipher whether LC axon degeneration is evident in post-mortem tissue from OBs of early AD cases, staged by Aβ and tau immunostainings (Thal-phase 1-2, Braak stage 1-2) and unaffected control brain donors. Strikingly, in the OB tissue from early AD cases, we revealed a pronounced degeneration of NET+ fibres compared to unaffected age-matched controls, which did not further decline in progressive AD cases (Fig. 6a–c). Moreover, we hypothesised that LC axon loss in humans, similar to mice, may correlate with an increased number of microglia. To this end, we performed TSPO-PET imaging in 16 patients with subjective cognitive decline (SCD)/ mild cognitive impairment (MCI), 16 AD patients and 14 unaffected controls, staged by Aβ and tau cerebrospinal fluid (CSF) levels, and investigated their TSPO signal in the respective OBs. We identified increased TSPO signals in the OBs of patients with prodromal AD, indicative of increased numbers or activation of microglia. Interestingly, even transitioning into AD diagnosis did not further elevate OB TSPO signals significantly (Fig. 6d, e). A number of independent longitudinal studies have highlighted olfactory deficits as a predictor of cognitive decline[7,8,44–46]. Thus, we analysed the data of our cohort for signs of hyposmia. While the prodromal AD group showed a trend towards olfactory deficits, patients transitioned into AD indeed revealed a significant decrease in the ability to identify common odours (Fig. 6f). Consequently, we asked whether these findings could be back-translated to $App^{NL-G-F}$

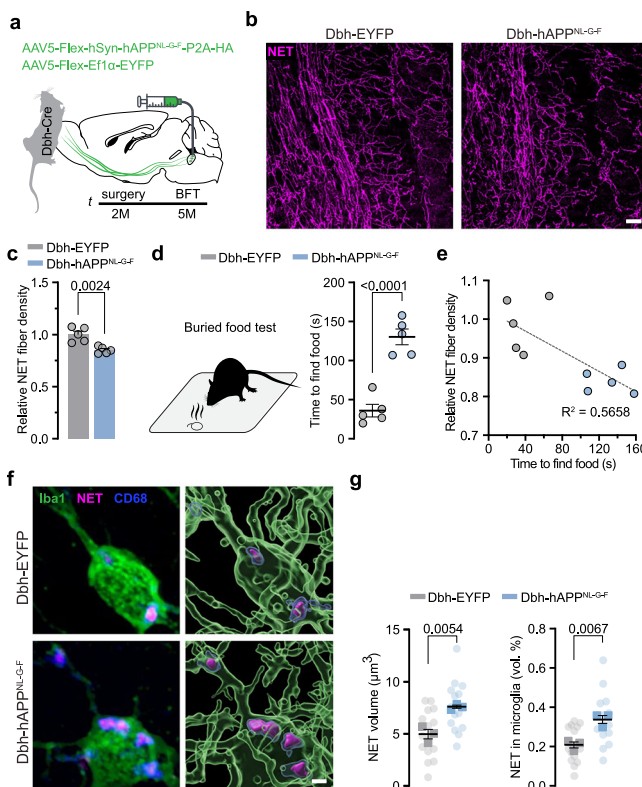

**Fig. 5 | LC specific *App^{NL-G-F}* expression causes OB LC axon degeneration and hyposmia. a** Experimental setup of *App^{NL-G-F}* virus injection into the LC of *Dbh-Cre* mice at 2 months of age. Graphic modified from Claudi, F. (2020). Mouse Top Detailed. Zenodo. https://doi.org/10.5281/zenodo.3925997. **b** Immunostaining of LC axons (NET, magenta) in the OB, 3 months post-injection. Scale bar: 50 μm. **c** Relative NET fibre density is reduced in *Dbh-hApp^{NL-G-F}* injected mice. (*Dbh-EYFP*: n = 5 vs. *Dbh-hApp^{NL-G-F}*: n = 5, each 3 slices). **d** Buried food test shows that *Dbh-hApp^{NL-G-F}* mice need more time to find the food pellet than *Dbh-EYFP* control injected mice. (*Dbh-EYFP*: n = 5 vs. *Dbh-hApp^{NL-G-F}*: n = 5). **e** Correlation between NET fibre density and time to find the buried food pellet. **f** Immunostaining and 3D reconstruction of microglia (Iba1, green), lysosomes (CD68, blue) and LC axon debris (NET, magenta). Scale bar: 2 μm. **g** Analysis of NET volume inside microglia. Dbh-*hApp^{NL-G-F}* microglia contain more NET+ signal than *Dbh-EYFP* microglia (*Dbh-EYFP*: n = 5 vs. *Dbh-hApp^{NL-G-F}*: n = 5, each 5 pictures); Data expressed as mean ± s.e.m.; **p < 0.01, ****p < 0.0001; Unpaired, two-tailed t-test in (**c, d, g**). Statistics shown in Supplementary Data 1. Source data are provided as a Source Data file.

mice. Indeed, TSPO-PET imaging in these animals revealed an early elevated signal in the OB compared to WT mice at 2-3 months of age, while the signal in the cortex of the same animals at that age remained unaltered, which was in line with previous reports[47] (Fig. 6g–j). Since elevated TSPO levels in mice can be either a sign of increased microglia density or activation, we analysed the TSPO expression on the single microglia cell level via co-immunostaining of Iba1 and TSPO and 3D reconstruction individual microglia from the OB of 3-months-old WT and *App^{NL-G-F}* mice. Indeed, we did not find significantly elevated TSPO expression in *App^{NL-G-F}* mice animals on the single microglia cell level, which suggest that elevated TSPO-PET signal reflects the increase in microglia density, rather than activated microglia (Supplementary Fig. 12). Of note, this not only supports our findings on microglia density with increasing age in *App^{NL-G-F}* mice, but also highlights the translatability of the mouse model, since elevated TSPO in humans has been shown to correlate with microglia density and not activation[47]. Thus, these translational data highlight and assign TSPO-PET imaging of the OB and hyposmia as a potential early biomarker of AD and LC-NA system dysfunction.

## Discussion

We reveal the LC-NA system degeneration as an impaired neuronal network to account for olfactory deficits in AD[10]. In humans, ~85% of AD patients exhibit early sensory deficits including hyposmia and anosmia, predicting cognitive decline[7–10,44–46]. Similarly, LC integrity is established as an early biomarker predicting cognitive decline in ageing and neurodegenerative diseases[42,43]. Interestingly, hyposmia is well documented in Parkinson's disease (PD), and LC dysfunction has been implicated to drive prodromal symptoms in PD. In contrast to the LC in AD, the OB and the dorsal motor nucleus of the vagus are the first sites to display α-synuclein pathology, likely suggesting an impairment of first-order olfactory neurons[48]. The well-established modulation of olfaction by LC-derived NA, especially in olfactory memory, underscores a possible link from LC vulnerability to hyposmia[49]. In our study, we detected LC axon loss in post-mortem OB tissue from prodromal AD patients. Notably, this pronounced early degeneration of LC axons did not progress further at later stages. However, using post-mortem tissue does not allow us to clarify the underlying mechanism of axon loss in humans, compared to *App^{NL-G-F}* mice. Neurodegeneration in the LC has been recently shown neuropathologically in post-mortem tissue along the AD continuum[50]. Whether LC axon degeneration in forebrain projection sites in humans precedes cell loss is not clear. A recent study indicates that at earlier stages (Braak 0-I), no cell loss in the LC can be detected, temporally aligning with our post-mortem tissue from human olfactory bulb, where LC axon density is reduced. Unfortunately, tissue from early AD stages is scarce, since patients typically decease at later stages of the disease or due to unrelated diseases such as heart attacks, which is usually not subject to brain donation. Similar to LC axon density, microgliosis detected by an elevated TSPO-PET signal in the OBs of SCD/MCI patients did not continue to increase in diagnosed AD patients. This likely reflects an increased number of microglia cells in the OB. Importantly, although in mice increased TSPO is often suggested to reflect activated microglia, we show that in the OB of *App^{NL-G-F}* mice, elevated TSPO levels, as in the human AD cohort indeed is likely driven by increased microglia density. These observations are in line with reports of increased number of microglia cells in the response to extracellular Aβ-oligomers and furthermore an increased microglia density in OB post-mortem tissue from AD patients[51]. This is reflected in the strong olfactory deficit of our AD patients, while we could only assign a slight trend to the prodromal AD group. Based on the substantial evidence of several independent studies that highlight hyposmia as a common early symptom in AD, we believe that this is likely due to our small cohort size[7,8,44–46,52–54]. The small size of our study cohort receiving TSPO-PET in conjunction with olfactory testing marks the limitation of our study. Further evaluation is needed to deduct a clear correlation between the TSPO-PET signal in the OB and the emergence of olfactory deficits in humans. In addition, the fast progress in MRI resolution and the sophisticated identification of the LC will enable a more detailed examination of the causal link between these two phenomena. Functional connectivity in live patients, together with resting-state activity, may then be able to delineate putative interconnections between these two widely separated anatomical regions. These in vivo imaging methods can also be combined with olfactory testing to establish a clear functional relationship. With sufficiently sized study cohorts, correlative data can be obtained to advance these variables to clinical testing. The fact that hyposmia and LC integrity are independent predictors of cognitive decline indicates that these processes may not only be correlating but may be causally linked. Indeed, early sophisticated work suggested that pharmacotoxic lesion of the LC exaggerates olfactory problems in APPPS1 mice, however, the experiments were conducted after nine months of consecutive toxin administration in 12-month-old animals[55]. Here, we provide a causal link between LC and olfactory deficits in mice. It is not clear which olfactory domain is impaired exactly, which clearly needs to be addressed in future

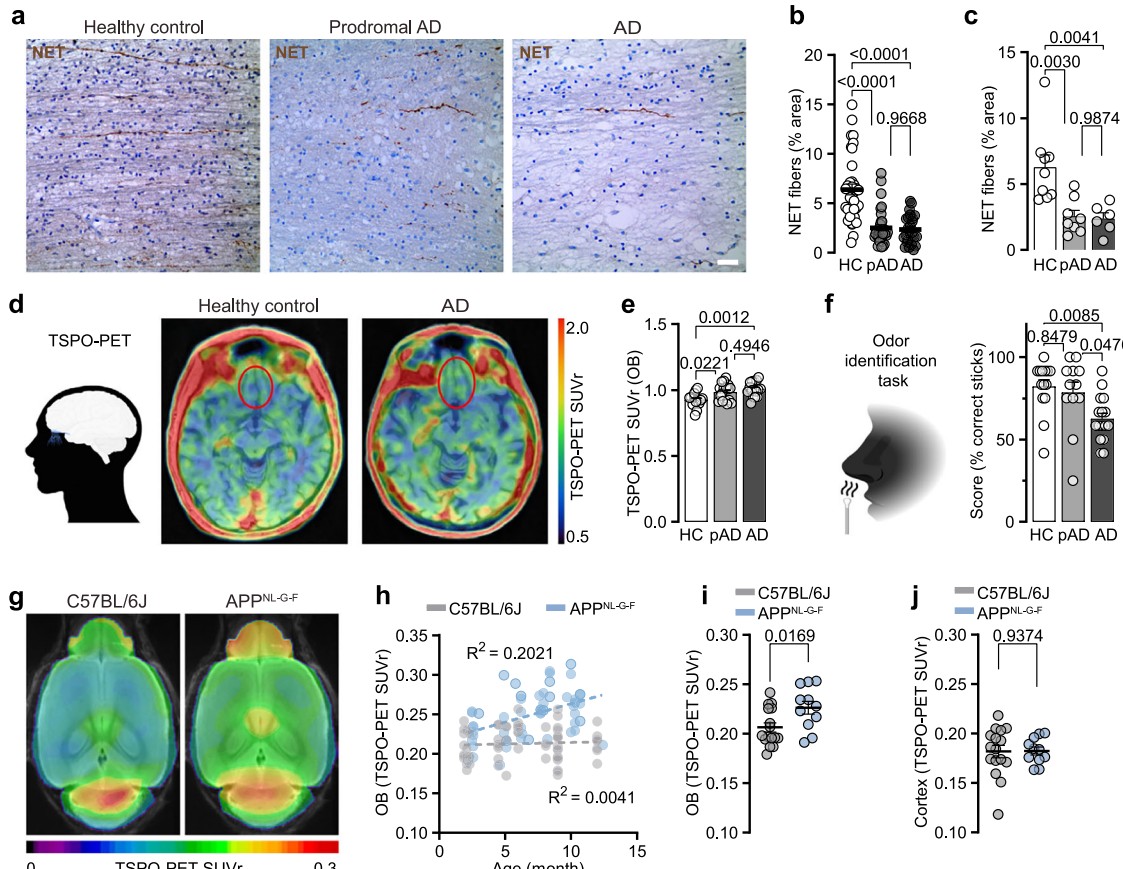

**Fig. 6 | TSPO-PET signals in mice and humans and LC axon loss in the OB of humans indicate hyposmia. a** Immunohistochemical staining of human OB brain sections stained for LC axons (NET, brown). Scale bar: 20 μm. **b** Quantification of percental NET fibre area per image and (HC: *n* = 36 vs. pAD: *n* = 32 vs. AD: *n* = 24) (**c**) per patient. (HC: *n* = 9 vs. pAD: *n* = 8 vs. AD: *n* = 6, each 4 pictures). **d** Schematic of OB in the human brain and a horizontal plane through the human brain, imaged with TSPO-PET. **e** Quantification of TSPO signal, comparing TSPO levels in unaffected brain donors, prodromal AD and AD patients (SUV: standardised uptake value). (HC: *n* = 14 vs. pAD: *n* = 17 vs. AD: *n* = 16). **f** Odour identification test in human participants shows the percental correct identification of odours, comparing unaffected patients with prodromal AD and AD patients. (HC: *n* = 14 vs. pAD: *n* = 12 vs. AD: *n* = 16). **g** Small-animal TSPO-PET in C57BL/6 J and *App*[NL-G-F] mice, horizontal plane through the brain at 3 months of age. **h** TSPO-PET signal in the OB, long-itudinally measured from 2 to 12 months of age. **i** At 2-3 months of age, *App*[NL-G-F] mice have a higher TSPO signal in the OB than C57BL/6 J mice, while (*n* = 16 C57BL/6 J vs. *n* = 11 *App*[NL-G-F]) (**j**) in the cortex no difference in TSPO signal was observed (*n* = 16 C57BL/6 J vs. *n* = 11 *App*[NL-G-F]); Data expressed as mean ± s.e.m.; ns, not significant; *p < 0.05, **p < 0.01, ****p < 0.0001; One-way ANOVA with Tukey's host-hoc test in (**b**, **c**, **e**, **f**); Unpaired, two-tailed t-test in (**i**, **j**); Statistics shown in Supplementary Data 1. Illustrations in 6 d and 6 f created in BioRender. Meyer, C. (2025) https://BioRender.com/ismo1ns, https://BioRender.com/1hib26h. Source data are provided as a Source Data file.

studies by deciphering olfactory decoding from odour representation in the glomeruli of the OB to more complex representations in higher olfactory centres. Since we describe a complex pattern of differential NA release upon odour stimulation, this will be particularly challenging in AD mouse models. OB activity changes need to be correlated to regions with differentially altered NA release using multi-colour in vivo 2 P imaging combined with odour stimulation. In addition, regions with similar responsiveness to specific odours need to be identified across animals and genotypes. Recent elegant work addressed the physiological role of NA modulation in the olfactory bulb[17]. In support of our findings, chemogenetic inhibition of NA release in the OB lead to similar increased times needed to find food in a buried food task. Collectively, the work suggests reduced odour sensitivity upon decreased NA, which might reflect in our acute slice mitral cell responses to NA.

LC dysfunction has classically been viewed as a consequence of tau pathology. It is considered to be the first region positive for hyperphosphorylated tau[56]. Due to this tau-centric view of LC dysfunction, the role of *App* and Aβ pathology in the LC in the aetiology of AD has only attracted little attention, although Aβ increases as a function of LC connectivity in rats[57]. In line, we provide evidence for an

*App* mutation-dependent axon loss underlying early olfactory deficits[56,58], marking the earliest described phenotype in this widely used AD mouse model to date. Of note, the degeneration was specific to NA axons, as neither ChAT[+] nor SERT[+] axon density declined early, but both are suggested to be affected early in disease progression[59]. Functionally, the pronounced reduction of NA-release in *App*[NL-G-F] mice upon odour stimulation can be considered a strong driver of the olfactory phenotype. With our cell-type-specific expression of *App*[NL-G-F] in LC neurons, we were able to demonstrate a coherent relationship between LC axon loss and olfactory deficits. However, the exact mechanism leading to LC axon loss in these experiments is not entirely clear. The downstream effects of *App* expression, unlike a transgenic humanised gene knock-in mouse (with physiological expression levels of *App*), may have other detrimental effects on neuronal function, such as impairing axonal transport[41,59]. Here, a recent paper specifically indicates an impairment of mitochondrial physiology and transport, both of which might underly a degeneration of the energy-demanding long, thin and unmyelinated axons of the LC. Intriguingly, the same work suggests a critical contribution of C-terminal fragments and Aβ-oligomers in the altered expression of transport proteins. Although not entirely clear, these data may support an LC intrinsic mechanism

resulting among others, in the hyperactivity observed by us and others[60]. Mechanistically, we present clear evidence that the expression of mutant human $App^{NL-G-F}$ instigates the externalisation of PS on LC axons. The $Ca^{2+}$-dependence of this externalisation is in line with the hyperactivity observed in our study. Moreover, similar AP frequency elevations have been recorded in *APPPS1* animals[36,60]. In the olfactory bulb, PS-dependent microglial phagocytosis plays a crucial role in both physiology and pathology. During development and adult neurogenesis, microglia mediate synaptic pruning via PS detection, which serves as a key mechanism to integrate newborn neurons into functional neuronal networks. Thus, PS located on hyperactive LC axons may be detected with a higher probability and fidelity compared to other regions. This provides a rationale for the early axon loss preceding all other highly LC-innervated regions, even in the absence of increased microglia-to-axon contact points. This is additionally reflected in the lack of an amyloid-driven DAM response in microglia extracted from the OB and the lack of changes in microglia contacts to NET⁺ axons. PS has recently been recognised as an opsonin in AD that marks neuronal structures for removal[27]. A variety of different receptors or effector-proteins subsequently trigger microglia-dependent clearance, including TREM-2 and MFG-E8. In line with the physiological role of PS-dependent microglia-driven synaptic remodelling, we reveal MFG-E8 as a mediator of microglia-dependent phagocytosis of LC axons. NA itself is a well-known modulator of microglia function, and decreased release from LC axons may have additional downstream effects on microglia[61–63]. Our data support the hypothesis that the OB is an anatomical region prone to detection of PS-MFG-E8 complexes by microglia, and thus axons of hyperactive LC neurons are cleared with a higher fidelity compared to other regions involving PS-MFG-E8-driven synaptic remodelling. In summary, we provide evidence for an underlying mechanism for hyposmia, an underappreciated sensory deficit in AD. Coordinated assessment of structural and functional connectivity, olfactory testing, together with CSF and blood biomarkers, could facilitate earlier AD diagnosis and be employed as solid predictors of disease progression and outcome. Ultimately, this may open the window for the earliest treatment to halt or decelerate disease progression.

## Methods

All animal experiments were approved by the Government of Upper Bavaria and followed the regulations of the Ludwig Maximilian University of Munich (ROB-55.2-2532.Vet_02-21-203). All human data was collected according to the guidelines of LMU and usage of human post-mortem material was approved by LMU ethical committee (23-0878).

### Animals

Mice, both male and female (1-6 months of age), were used and held on a 12 h light/dark cycle with food and water ad libitum. The $App^{NL-G-F}$ mouse line is a knock-in model, were pathogenic Aβ is elevated by inserting 3 different mutations associated with AD[41]. Crossing $App^{NL-G-F}$ mice with *Dbh*-Cre was used to manipulate the locus coeruleus-noradrenergic system. *Dbh*-Cre mice express the Cre recombinase under the *dbh* (dopamine beta hydroxylase) promotor[64]. $App^{NL-G-F}$ mice were also crossed with TSPO-KO[33] mice to access the effect of a TSPO knock-out on the noradrenergic system. All animals were maintained on a C57BL/6 J background, and as control animals, C57BL/6 J mice were used, purchased from the Jackson Laboratory (Maine, United States).

### Immunostaining: Mouse brain tissue

Mice were deeply anaesthetised and transcardially perfused with phosphate-buffered saline (PBS) and 4% paraformaldehyde (PFA). Brains got fixed by immersion in PFA at 4 °C for 16 h. 50 μm thick slices were cut in a coronal plane using a vibratome (VT1200S, Leica Biosystems). Each 4 slices per animal containing the olfactory bulb, piriform cortex, hippocampus and locus coeruleus were used for an

immunostaining analysis. Staining was performed on free-floating sections. Slices were blocked with blocking solution (10% normal goat serum and 10% normal donkey serum in 0.3%Triton and PBS) for 2 h at RT. Primary antibodies were incubated overnight at 4 °C, followed by washing and secondary antibody incubation for 2 h at RT, protected against light. Slices were mounted and cover-slipped with mounting medium, containing DAPI (Dako, Santa Clara, USA). Primary antibodies used were: rabbit anti-NET (1:500, Abcam, ab254361), mouse anti-NET (1:1000, Thermo Fisher, MA5-24547), guinea pig anti-Iba1 (1:500, Synaptic Systems, 234308), chicken anti-TH (1:1000, Abcam, ab76442), mouse anti-Aβ (NAB228) (1:500, Santa Cruz, sc-3277), rat anti-CD68 (1:500, BioRad, MCA1957), goat anti-MFG-E8 (1:500, R&D Systems, AF2805), rabbit anti-C1q (1:1000, Abcam, ab182451), chicken anti-GFP (1:1000, Abcam, ab13970), rabbit anti-GFP (1:1000, Thermo Fisher, A21311), rabbit, HA-tag (1:500, Sigma, H6908), Streptavidin 488 (1:1000, Invitrogen, S32354), Streptavidin 647 (1:1000, Invitrogen, S32357).

### Image acquisition

Three-dimensional images were acquired with a Zeiss LSM900 confocal microscope (Carl Zeiss, Oberkochen).

### NET fibre quantification

For the quantification of the NET fibre density as well as Iba1-microglia and NAB288-Aβ-plaque area, a 10x objective (8-bit stacks of 101.41 μm x 101.41 μm x 25 μm) was used. The staining density (area %) was analysed with ImageJ. After a manual brightness/contrast adjustment, a threshold was set to calculate the perceptual area of NET-positive LC fibres, Iba1-positive microglia and NAB288-positive Aβ plaques. Results from 4 sections per animal from 4–8 animals per group were averaged and reported as mean ± s.e.m.

### Colocalization analysis

For the engulfment of NET in microglia, airyscan images were taken with a 63 x/1.4 x NA oil immersion objective. Z-stack images were acquired of 8 microglia per mouse from 3 animals per group in the external plexiform layer, covering 30 μm at 0.14 μm intervals. Colocalization of Iba1⁺ microglia - NET⁺ LC axon contact points was analysed on 15 μm z-stack images (40 x/1.3 x magnification, 0.3 μm intervals) of 6 pictures per mouse, 3 mice per genotype. Colocalization of PS on NET⁺ LC axon was analysed on 15 μm z-stack images (40 x/0.7 x magnification, 0.3 μm intervals) of 7 pictures per mouse, 3 mice per genotype. Colocalization of C1q on NET⁺ LC axon was analysed on 6 μm z-stack images (63 x/1.4 x magnification, 0.18 μm intervals) of 5 pictures per mouse, 2 mice per genotype. Colocalization of MFG-E8 on NET⁺ LC axon was analysed on 15 μm z-stack images (40 x/0.7 x magnification, 0.3 μm intervals) of 6 pictures per mouse, 4 mice per genotype. Colocalization of TSPO in Iba1⁺ microglia was analysed on 20 μm z-stack image (40 x magnification, 0.22 μm intervals) of 5-6 regions per animal, 3 mice per genotype. All images were 3-D reconstruction in IMARIS (Bitplane, 9.6.1) using the Surface module. Colocalization was measured in volume and normalised to the NET axon density.

### Staining: Human brain tissue

Human brain tissue from 9 healthy unaffected brain donors, 8 prodromal AD subjects and 6 AD patients was provided from the Munich brain bank. Demographic details of the subjects are listed in Supplementary Data 3. Paraffin-embedded brain sections (5 μm) of the olfactory bulb were cut in a horizontal plane, using a microtome (Leica SM2010R) and mounted on glass slides until further processing. Sections were deparaffinized with xylene and rehydrated through a series of descending alcohol concentrations. For the DAB staining, an automated IHC/SH slice staining system (Ventana BenchMark ULTRA) was used. On separate slices, NET 1:200, Aβ 1:5000 and Tau 1:400 was stained and visualised with an upright Bridgefield microscope. Each 4

pictures per subject (20 x magnification) were acquired and analysed regarding their perceptual density of NET⁺ LC axons.

## Microglia isolation

Primary microglia were isolated from the olfactory bulb of 2-month-old C57BL/6 J and $App^{NL-G-F}$ mice using MACS technology (Miltenyi Biotec) according to the manufacturer's instructions. Briefly, mice were perfused with PBS and the brain washed in ice-cold HBSS (Gibco) supplemented with 7 mM HEPES (Gibco). Chopped tissue pieces were incubated with digestion medium D-MEM/GlutaMax high glucose and pyruvate (Gibco) supplemented with 20 U papain per ml (Sigma P3125) and 0.01% L-Cysteine (Sigma) for 15 min at 37 °C in a water bath. Subsequently, enzymatic digestion was stopped using a blocking medium, 10% heat-inactivated FBS (Sigma) in D-MEMGlutaMax high glucose and pyruvate. Mechanical dissociation was gently but thoroughly performed by using three fire-polished, BSA-coated glass Pasteur pipettes with decreasing diameter. Subsequently, microglia were magnetically labelled with CD11b microbeads (Miltenyi Biotec, 130-097-678) in MACS buffer (0.5% BSA, 2 mM EDTA in 1 x PBS, sterile filtered) and the suspension loaded onto a pre-washed LS-column (Miltenyi Biotec, 130-042-401). Following washing with 3 × 1 ml MACS buffer, magnetic separation resulted in a CD11b enriched and a CD11b depleted fraction. To increase purity further, the microglia-enriched fraction was loaded onto another LS-column. The total numbers of obtained microglia fractions were quantified using C-Chip chambers (Nano EnTek, DHC-N01). Isolated primary microglia were washed twice with 1 x PBS (Gibco) and immediately processed for sequencing or plated for a phagocytosis assay.

## Phagocytosis assay

Synaptic Protein was enriched using the Syn-PER™ Synaptic Protein Extraction Reagent (Thermo Fisher) according to the manufacturer's protocol and published previously[65]. In brief, fresh brains from C57BL/6 J mice at 4 months of age were isolated and homogenised in 10 mL/g of brain tissue of Syn-PER™ reagent substituted with protease and phosphatase inhibitor. The homogenate was then centrifuged at 1200 x g at 4 °C for 10 min. The supernatant containing the synaptic fraction was then transferred into a new tube and spun at 15.000 x g at 4 °C for 20 min. The supernatant was aspirated, and the pellet of synaptic protein was resuspended in 1 mL of Syn-PER™ reagent containing 5% (v/v) DMSO per gram tissue originally used. Synaptosome extracts were then stored at −80 °C before further usage. Synaptic Protein was labelled with the pHrodo™ Red succinimidyl ester (Thermo Fisher Scientific), which emits a red fluorescent signal only in acidic environments. Labelling was performed as previously described[66]. In brief, synaptic protein was washed in 100 mM sodium bicarbonate, pH 8.5 and spun down (17,000 x g for 4 min at 4 °C). pHrodo™ dye was dissolved in 150 μL DMSO per 1 mg dye to a concentration of 10 mM. The pHrodo™ stock solution was added to the synaptic protein at a concentration 1 μl pHrodo per 1 mg of synaptic protein. After incubating at room temperature for 2 h, protected from light, the labelled protein was washed twice in DPBS and spun down (at 17,000 x g for 4 min at 4 °C). After resuspending synaptic protein with 100 mM sodium bicarbonate, pH 8.5, to a concentration of 1000 μg/ml, it was aliquoted and stored at −80 °C before usage. Primary microglia were cultured in tissue culture-treated 96-well plates in microglia-medium, adding freshly 10 ng/ml GM-CSF (R&D Systems) for three days in vitro (DIV) at 37 °C, 5% CO2, changing medium at DIV 1. For the phagocytic uptake assay, medium was replaced with medium in which pHrodo™ labelled synaptic protein was resuspended at the desired concentration (2.5 μg/mL). For the Cytochalasin D (CytoD) control, cells were treated with 10 μM CytoD (Sigma) for 30 min, before adding medium with labelled synaptic protein and CytoD. Immediately after adding the substrates, the cells were placed in an Incucyte™ S3 Live-Cell Analysis System (Sartorius). Scans were performed every hour

with 20 x magnification and both phase contrast and red fluorescent channels, acquiring a minimum of three images per well and scan. Quantification was done using the cell-by-cell adherent analysis. Phagocytic index was calculated using the total integrated intensity (RCU x μm²/Image) normalised to the number of cells per image.

## NA ELISA

In order to measure the potential difference in the noradrenaline concentration between C57BL/6 J mice and $App^{NL-G-F}$ mice, a noradrenaline ELISA was carried out. Mice were deeply anaesthetised and perfused with PBS, and their brains rapidly removed. The olfactory bulb was dissected and snap frozen using liquid nitrogen. The tissue was homogenised in 0.01 M HCl in the presence of 0.15 mM EDTA and 4 mM sodium metabisulfite, before being processed with an ELISA kit (BA E-5200) according to the manufacturer's protocol.

## RNA sequencing and Bioinformatics

RNA was isolated from microglial cell pellets using the RNeasy Plus Micro kit (Qiagen, 74034). Briefly, samples were lysed with RLT Plus lysis buffer containing beta-Mercaptoethanol, genomic DNA was removed by passing the lysate through gDNA eliminator columns, and the eluate was applied to RNeasy spin columns. Contaminants were removed with repeated Ethanol washes before RNA was eluted with 20 μL molecular-grade water. All steps were carried out automatically on a Qiacube machine. RNA was quantified on a Qubit Fluorometer (Invitrogen, Q33230) and 6 ng of total RNA were used as input for library preparation with the Takara SMART-seq Stranded kit (Takara, 634444) following the manufacturer's instructions. Fragmentation time was kept at 6 min, and AMPure XP beads (Beckman Coulter, A63880) were used for all clean-up steps. Library QC using a Bioanalyzer revealed average insert sizes around 350 bps. The molarity of each of the 16 libraries was determined by using the ddPCR Library Quantification Kit for Illumina TruSeq (Bio-Rad, 1863040) according to the manufacturer's instructions. Libraries were then diluted to 4 nM and pooled in an equimolar fashion. Paired-end sequencing was carried out for 150 cycles on a NextSeq 550 sequencer (Illumina, 20024907) using a High-Output flow cell. After sample demultiplexing, reads were aligned using STAR v2.7.8 to a customised genome based on the GRCm39 assembly and the gencode vM32 primary annotation that additionally contained sequences and annotations for the human App gene. Group assignments were verified by manually inspecting alignments to the (human) App sequence and checking for the presence of the NL-, G- and F- mutations in transgenic animals. The count matrix produced by STAR v2.7.8 was used as an input for differential expression testing using edgeR. The count matrix was filtered to retain genes with at least 5 counts in at least 50% of samples, and quasi-likelihood tests were conducted after fitting appropriate binomial models. Differential expression was considered significant if FDR < 0.1 and if the absolute log-fold-change exceeded 0.5. Gene lists were annotated with the enrichR package. All analyses made heavy use of the tidyverse and ggplot2 packages and were performed on a server running Arch Linux, R version 4.3.2 and RStudio Server 2023.03.0.

## Behavioural olfactory tests

All behavioural experiments were conducted during the light phase of the animals and were performed in a blinded manner. To evaluate possible differences in odour performance, C57BL/6 J and $App^{NL-G-F}$ mice at 1, 3 and 6 months of age underwent a buried food test. One day before the test, animals got food-deprived for 18 h. On the test day, animals got acclimated to the new environment for at least 30 min in a fresh cage with increased bedding volume. The test begins with placing the animal in the test cage with a food pellet buried in the bedding. The time it takes for the animals to reach the food pellet was analysed based on a video recording. The mean search time that the two groups took to find the food pellet was calculated and compared

by an unpaired Student's *t*test. The sensitivity test evaluates whether mice can perceive odours even at weak concentrations. At the beginning of the experiment, the animals got acclimated to the odour applicator (a dry cotton swab without odour) for 30 min to exclude the applicator itself as a potential source of error and a new, interesting object. For the test, a pleasant-smelling odour "vanilla" got applied to a cotton swab in two ascending concentrations (1:1000 and 1:1 in water), and each concentration got presented to the mouse for 2 min consecutively, with 1 min break in between to change the odorant. Water, in which all odours are dissolved, was used as a control. Mice were filmed from the top and side with 2 synchronised cameras, and their nose was segmented and tracked offline in both videos using 2 custom-trained S.L.E.A.P. networks[67]. A custom Python code was used to track the 3D position of the nose relative to the odour-dispersing cotton tip, and to quantify the time spent interacting with the different odour concentrations (investigation zone < 2 cm nose to cotton tip).

## Virus injections

Different viral injections into the LC region or the olfactory bulb were carried out in this study. For injections into the olfactory bulb the following coordinates were used: right OB (AP: 5.00, ML: −1.07, DV: 2.57) and left OB (AP: 4.28, ML: 0.41, DV: 2.45), while injection into the LC region were made using the following coordinate: left LC (AP: −5.44, ML: − 0.89, DV: 4.07) and right LC (AP:−5.44, ML: −0.99, DV: 3.99). Adjustments were made if blood vessels were right on top of the injection location. AAV-hSyn-DIO-h3MDGs / AAV1-Syn-GCamp8f; Chemogenetic activation of LC neurons was carried out to investigate if an increase in noradrenaline release could rescue the impaired olfaction in $App^{NL-G-F}$ x *Dbh*-Cre mice. 5-month-old mice were bilaterally injected in the LC with AAV-hSyn-DIO-h3MDGs or the control AAV1-Syn-GCamp8f. To activate H3MDGs 1 month post-injection, mice were injected i.p. with 1 mg/kg CNO 30 min before undergoing the buried food test. For patch clamp recordings, a concentration of 3 μM was used. AAV5-Flex-hSyn1-APP$^{NL-G-F}$-P2A-HA / AAV-5-Flex-Ef1α-EYFP; To investigate APP$^{NL-G-F}$ expression exclusively in the LC, we designed a custom-build Cre-dependent AAV virus. It is a mammalian FLEX conditional gene expression AAV virus (Cre-on) with the full vector name: pAAV[FLEXon]-SYN1 > LL:rev({hAPP(KM670/671NL,I716F)}/P2A/HA):rev(LL):WPRE (Vector ID: VB230525-1787fff). The virus is flagged with an HA-tag for post-hoc virus expression validation.

## Chronic olfactory bulb window implantation

To study pathology-dependent norepinephrine release in the olfactory bulb, 2-month-old $App^{NL-G-F}$ mice (*n* = 3) and C57BL/6 J (*n* = 3) control animals were fitted with cranial windows. In short, mice were anaesthetised with a mixture of Medetomidin, Midazolam and Fentanyl at 0.5, 5 and 0.05 mg/kg bodyweight, respectively. Dexamethasone was injected i.p. at 100 mg/kg to reduce inflammatory responses, and the animal got head-fixed in a stereotactic frame. The skin was cut vertically to expose lambda, bregma and the olfactory bulb and give adequate adherence space for the headbar. Surface edging was performed by scoring the skull lightly with a scalpel and applying a UV light curing mildly corrosive agent (IBond Self Etch, Kulzer 66046243). After locating the rostral rhinal vein, running just posterior of the olfactory bulb, a 3 mm biopsy punch was used to indicate the craniotomy location just anterior of the vein. The Neurostar surgical robot was the used to drill the marked circle until the skull disk could be removed. The dura mater was removed on the exposed part of the left olfactory bulb. The norepinephrine sensor pAAV-hSyn-GRAB-NE1m or the mutated version pAAV-hSyn-GRAB-NE$_{mut}$ was injected into the centre of the bulb (450 nl at 45 nl/min) at a depth of 400 μm. After the injection, the area was cleaned, and a 3 mm circular cover slip fitted over the craniotomy area. The window was fixed in place with tissue adhesive glue (Surgibond tissue adhesive, Praxisdienst, 190740). The entire area with exposed skull was subsequently filled with dental

cement (Gradia Direct Flo BW, Spree Dental, 2485494) and a headbar suitable for the later utilised 2P-microscope quickly placed over the window. The cement was cured with UV. After surgery, the mice received 5 mg/kg Enrofloxacin as an antibiotic, 25 mg/kg Carprofen to reduce inflammation and 0.1 mg/kg Buprenorphin as an analgesic. A mixture of Atipamezol and Flumazenil (2.5 and 0.5 mg/kg) was used to antagonise the anaesthesia. In total, 3 WT and 3 $App^{NL-G-F}$ mice were used in the first round with odour trials for banana and lemon, while the second round with odours banana and caraway consisted of 4 WT and 5 $App^{NL-G-F}$ animals. Three animals were used as control injected with the mutated version of GRAB$_{NE}$.

## Two-photon imaging

One month after surgery, all mice were trained on the wheel used for awake in vivo imaging, their windows cleaned, and the injection site checked for expression. A delivery method for a vanilla scent was established by combining a tube connected to a picospritzer system (PSES-02DX) with a vial containing vanilla aroma (Butter-Vanille, Dr. Oetker, 60-1-01-144800), lemon aroma (Natürliches Zitronen Aroma, Dr. Oetker, 1-46-112100), banana aroma (1-Hexanol, Sigma Aldrich H-13303), caraway aroma ((S)-(+)-carvone, Sigma Aldrich 22070) or mineral oil (Sigma Aldrich 330779). For $10^4$ dilution studies, banana and caraway aroma were diluted 1:10000 in mineral oil and compared recordings with $10^0$ undiluted samples. The tube opening was placed at a fixed distance of roughly 4 cm in front of the mouse, and a vacuum pump placed slightly behind the head to ensure quick dispersion of the scent after an air puff was delivered. The two-photon microscope system was the Femtonics ATLAS system with a Coherent Chameleon tunable laser set at 920 nm. Three locations (field of views; FOV) were imaged per mouse and odour at depths between 30 and 60 μm below the surface with a 16x objective. For concentration comparisons, the same locations were rediscovered to achieve comparability. Over three minute,s a z-stack of 120 x 120 x 30 μm with a pixel size of 0.22 μm and a z step of 1 μm was recorded at 1.13 Hz. After one minute of baseline recording, 10 s of an odour-delivering airpuff were administered. After each three-minute recording, 20 min of waiting time separated the subsequent recording and ensured the dispersion of the odour inside of the imaging setup. For an additional long-term trial, one WT mouse was imaged for 18 min with the above mentioned settings. Here, vanilla airpuffs at 10 s of length were applied at 5, 10 and 15 min.

## Analysis of 2-Photon imaging

All recordings were loaded into Fiji, and each z-stack projected with a summation of all 30 z-slices. Afterwards, the EZCalcium Motion Correction (based on NoRMCorre) (PMID: 32499682) was used to reduce motion artefacts. For each individual recording, the frame brightness was normalised to the average of the baseline frames 20–67 before the air puff (frame 68) and the average of the three adjusted curves calculated. The first 20 frames were removed to account for inconsistencies at the start of each recording, such as startling of the animal. After considering different windows for comparison, an analysis of all odours at $10^0$ lead to the selection of frame 89-91 (Supplementary Fig. 3c). Each recording (FOV) was divided into 36 subtiles (ROIs). For animal averages, first, the 3 brightest ROIs at baseline of each FOV were chosen to avoid blood vessel expansion as a contributor to the ΔF/F values obtained. After averaging these three ROIs, the resulting three values were averaged again. To gain more clarity in signal composition, we also show all 36 (ROIs) x 3 (FOVS) values for each animal and their respective increase or decrease in the chosen analysis window. For the 18 min recording, the average was taken from frames 20–300 for normalisation. Heatmaps were created with the Python Seaborn distribution.

## Acute slice electrophysiology (perforated-patch-clamp)

Acute brain slice recordings were performed as previously described[68–70]. Mice were anaesthetised with isoflurane and

subsequently decapitated, before the brain was rapidly removed and stored in cold (4 °C) glycerol aCSF. 300 μm thick slices containing the region of the locus coeruleus and the olfactory bulb were cut in carbogenated (95% O2 and 5% CO2) glycerol aCSF (230 mM Glycerol, 2.5 mM KCl, 1.2 mM NaH2PO4, 10 mM HEPES, 21 mM NaHCO3, 5 mM glucose, 2 mM MgCl2, 2 mM CaCl2 (pH 7.2, 300-310 mOsm), using a vibration microtome (Leica VT1200S, Leica Biosystems, Wetzlar, Germany). Slices were immediately transferred into a maintenance chamber with warm (36 °C) carbogenated aCSF (125 mM NaCl, 2.5 mM KCl, 1.2 mM NaH2PO4, 10 mM HEPES, 21 mM NaHCO3, 5 mM glucose, 2 mM MgCl2, 2 mM CaCl2 (pH 7.2, 300-310 mOsm)). After 50 min recovery, slices were kept at room temperature (-22 °C) waiting for recordings. For electrophysiological recordings, slices were individually transferred into a recording chamber and perfused with carbogenated aCSF at a flow rate of 2.5 ml/min. The temperature was controlled with a heat controller and set to 26 °C. Perforated patch-clamp recordings were obtained from LC neurons and OB mitral cells visualised with an upright microscope, using a 60x water immersion objective. Biocytin labelling and post-hoc immunohistochemistry was used to confirm the right cell type. Patch pipettes were fabricated from borosilicate glass capillaries (outer diameter: 1.5 mm, inner diameter: 0.86 mm, length: 100 mm, Harvard Apparatus) with a vertical pipette puller (Narishige PC-10, Narishige Int. Ltd., London, UK). When filled with internal solution (tip-filled with potassium-D-gluconate intracellular pipette solution 1: 140 mM potassium-D-gluconate, 10 mM KCl, 10 mM HEPES, 0.1 mM EGTA, 2 mM MgCl2 (pH 7.2, ~290 mOsm) and back-filled with potassium-D-gluconate intracellular pipette solution 2: 140 mM potassium-D-gluconate, 10 mM KCl, 10 mM HEPES, 0.1 mM EGTA, 2 mM MgCl2, 0.02% Rhodamine Dextran, ~200 mg/ml Amphotericin B (dissolved in DMSO) and if needed 1% biocytin (pH 7.2, ~290 mOsm), they had a resistance of 4-5 MOhm. All experiments were performed using an EPC10 patch clamp (HEKA, Lambrecht, Germany) and controlled with the software PatchMaster (version 2.32; HEKA). The liquid junction potential (-14.6 mV) was compensated prior to seal formation, and recordings were always compensated for series resistance and capacity. All executed protocols were recorded with Spike 2 (version 10a, Cambridge Electronic Design, Cambridge, UK). Data were sampled with 10 to 25 kHz and low-pass filtered with a 2 kHz Bessel filter.

### Human TSPO-PET imaging acquisition and analysis

For PET imaging an established standardised protocol was used[71–73]. All participants were scanned at the Department of Nuclear Medicine, LMU Munich, using a Biograph 64 PET/CT scanner (Siemens, Erlangen, Germany). Before each PET acquisition, a low-dose CT scan was performed for attenuation correction. Emission data of TSPO-PET were acquired from 60 to 80 min after the injection of 187 ± 11 MBq [18F]GE-180 as an intravenous bolus, with some patients receiving dynamic PET imaging over 90 min. The specific activity was > 1500 GBq/μmol at the end of radiosynthesis, and the injected mass was 0.13 ± 0.05 nmol. All participants provided written informed consent before the PET scans. Images were consistently reconstructed using a 3-dimensional ordered subsets expectation maximisation algorithm (16 iterations, 4 subsets, 4 mm Gaussian filter) with a matrix size of 336 × 336 × 109, and a voxel size of 1.018 × 1.018 × 2.027 mm. Standard corrections for attenuation, scatter, decay, and random counts were applied. The 60–80 min p.i. images of all patients and controls were analysed.

### Small animal TSPO μPET

All small animal positron emission tomography (μPET) procedures followed an established standardised protocol for radiochemistry, acquisition and post-processing[74,75]. In brief, [18F]GE-180 TSPO μPET with an emission window of 60–0 mins post-injection was used to measure cerebral microglial activity. $App^{NL-G-F}$ and age-matched C57BL/6 mice were studied at ages between two and twelve months. The TSPO μPET signal in the cortex and the hippocampus was previously reported in other studies[76–78]. All analyses were performed by PMOD (V3.5, PMOD technologies, Basel, Switzerland). Normalisation of injected activity was performed by the previously validated myocardium correction method[79]. TSPO μPET estimates deriving from predefined volumes of interest of the Mirrione atlas[80] were used: olfactory bulb (22.9 ± 1.5 mm³) and cortical composite (144.9 ± 6.0 mm³). Associations of TSPO μPET estimates with age and genotype, as well as the interaction of age*genotype were tested by a linear regression model. We performed all PET data analyses using PMOD (V3.9; PMOD Technologies LLC; Zurich; Switzerland). The primary analysis used static emission recordings, which were coregistered to the Montreal Neurology Institute (MNI) space using non-linear warping (16 iterations, frequency cutoff 25, transient input smoothing 8 × 8 x 8 mm³) to a tracer-specific template acquired in previous in-house studies. Intensity normalisation of all PET images was performed by calculation of standardised uptake value ratios (SUVr) using the cerebellum as an established pseudo-reference tissue for TSPO-PET[81].

### Human olfactory test

For detecting decreased olfactory performance due to neurodegenerative diseases, the "Sniffin' Sticks - Screening 12" test was employed. Developed in collaboration with the Working Group "Olfactology and Gustology" of the German Society for Otorhinolaryngology, Head and Neck Surgery, the test provides a preliminary diagnostic orientation and can be conveniently used in everyday settings. It classifies individuals as anosmics (no olfactory ability), hyposmics (reduced olfactory ability), or normosmics (normal olfactory ability)[82]. The participants are presented with 12 familiar scents (health-safe aromas, mostly used in food as flavourings) separately, in succession. Both nostrils are assessed simultaneously. Each scent is presented with a multiple-choice format, where participants choose one of four terms that best describe the scent, even if they perceive no smell. During testing, no feedback is provided to ensure unbiased responses. Demographic details of the subjects are listed in Supplementary Data 3.

### Statistics and reproducibility

Statistical details of every experiment, including the number of technical and biological replicates are explained in Supplementary Data 1 and 2. No statistical method was used to predetermine sample size. Excluded data is mentioned in the Reporting Summary. Where possible, the Investigators were blinded to allocation during experiments and data analysis. All statistical analyses were performed in GraphPad Prism (version 10.1.1). Data are reported as mean ± s.e.m. Significance was set at $P < 0.05$ and expressed as *$P < 0.05$, **$P < 0.01$, ***$P < 0.001$ and ****$P < 0.0001$.

### Reporting summary

Further information on research design is available in the Nature Portfolio Reporting Summary linked to this article.

## Data availability

All data sets generated in this study for all figures are available through the corresponding Source data files. Source data are provided with this paper. Transcriptomic data is deposited at Gene Expression Omnibus (https://www.ncbi.nlm.nih.gov/geo/) under the GEO submission ID: GSE302245. The raw data sets and any further information for the reanalysis of data reported in this paper will be made available from the lead contact (lars.paeger@dzne.de) upon request. There are no restrictions to the data availability. Source data are provided in this paper.

## Code availability

Code for the analysis of the OB microglia transcriptome is available at GitHub (https://github.com/fstrueb/OB_APPKI_RNAseq). Any other

code generated for analysis can be recapitulated from information in the Methods section and is available from the lead contact upon request. There are no restrictions on code availability.

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

## Acknowledgements

We thank T. Saito and T. C. Saido for providing the *App^NL-G-F* mice. Special thanks goes to Fang Zhang, Michael Schmidt, Anke Jürgensonn, Marcel Matt and Brigitte Haslbeck for outstanding technical and administrative assistance; the whole staff, especially Ekrem Göcmenoglu, of the animal facility at the Centre for Stroke and Dementia under the lead of Dr. Anne van Thaden, Dr. Carolin Ildiko Konrad and Dr. Manuela Schneider for their continuous support on all animal related efforts. We thank Prof. Dr. Neville Vassallo for vivid discussions and manuscript revision. We kindly thank SciDraw (https://scidraw.io, license CC-By 4.0) and BioRender (https://www.biorender.com/) for providing the images used (orginal or modified) in our figures (https://doi.org/10.5281/zenodo.3926119, https://doi.org/10.5281/zenodo.3925903, https://doi.org/10.5281/zenodo.3925997, https://doi.org/10.5281/zenodo.3925917, https://doi.org/10.5281/zenodo.3926033, https://doi.org/10.5281/zenodo.3925987), (Fig. 6d: Created in BioRender. Meyer, C. (2025) https://BioRender.com/ismo1ns, Fig. 6f: Created in BioRender. Meyer, C. (2025)

https://BioRender.com/1hib26h). This work was partly supported by the Deutsche Forschungsgemeinschaft (DFG, German Research Foundation) under DFG Research Unit FOR 2858 (project number 403161218; applies to L.P., M.B., S.T., J.H.), DFG Priority Programme SPP2395 (TA 551/2-1; applies to S.T.) and under Germany's Excellence Strategy within the framework of the Munich Cluster for Systems Neurology (EXC 2145 SyNergy– ID 390857198; applies to R.P., M.B., J.J.N. and J.H.). R.P. is supported by the German Centre for Neurodegenerative Diseases (Deutsches Zentrum für Neurodegenerative Erkrankungen, DZNE), the Davos Alzheimer's Collaborative, the VERUM Foundation, the Robert-Vogel-Foundation, the National Institute for Health and Care Research (NIHR) Sheffield Biomedical Research Centre (NIHR203321), the University of Cambridge and the Ludwig-Maximilians-University Munich Strategic Partnership within the framework of the German Excellence Initiative and Excellence Strategy and the European Commission under the Innovative Health Initiative programme (project 101132356).

## Author contributions

C.M. designed and conducted most experiments (electrophysiological recordings, immunohistological staining, olfactory behaviour tests, imaging of mouse and human brain tissue, ELISA, data analysis and 3D reconstruction) and performed manuscript preparation. T.N. performed olfactory bulb window surgery and NA 2-photon in vivo measurements. P.F. performed olfactory bulb window surgery and NA 2-photon in vivo measurements and analysed olfactory sensitivity tests. F.S. performed olfactory bulb microglia sequencing and subsequent data analysis. N.F.L. performed chemogenetic experiments. B.R. performed human odour identification tests and subsequent analysis. J.G. performed immunofluorescent staining, confocal imaging and 3D reconstruction. K.K. performed immunofluorescent staining, confocal imaging and 3D reconstruction. S.L.R. performed immunofluorescent staining and confocal imaging. Y.T. performed microglia isolation, phagocytosis assay and subsequent analysis. D.P. assisted with the maintenance of the experimental mice and performed mouse tissue preparation. Y.S. assisted with the maintenance of the experimental mice and performed mouse tissue preparation. G.M.K. assisted with the preparation of the ethical approval and maintenance of experimental mice. K.O. performed microglia isolation for sequencing. K.W. performed a small animal PET study and analysis. G.B. performed a small animal PET study and analysis. J.W. helped to establish the MFG-E8 antibody stain. S.G. performed human odour identification tests and subsequent analysis. C.K. performed human odour identification tests and subsequent analysis. M.S. performed human odour identification tests and subsequent analysis. R.B.B. provided TSPO-KO mice. G.L. provided TSPO-KO mice. R.J.M. provided TSPO-KO mice. R.P. performed a human PET study and analysis. T.K. performed project planning. J.J.N. performed project planning. S.T. performed project planning. M.B. performed a small animal PET study and analysis. J.H. performed project planning and manuscript revision. L.P. performed virus injections, project planning and supervision and wrote the manuscript with input from all authors. All authors provided comments and approved the manuscript.

## Funding

## Competing interests

M.B. received consulting/speaker honoraria from Life Molecular Imaging, GE Healthcare, and Roche, and reader honoraria from Life Molecular Imaging. All other authors declare no competing interests.

## Additional information

[1]German Center for Neurodegenerative Diseases (DZNE), Munich, Germany. [2]Center for Neuropathology and Prion Research, Ludwig-Maximilians-Universität, Munich, Germany. [3]Institute of Neuroradiology, LMU Hospital, LMU Munich, Munich, Germany. [4]Department of Psychiatry and Psychotherapy, LMU Hospital, LMU Munich, Munich, Germany. [5]Department of Nuclear Medicine, University Hospital of Munich, Ludwig-Maximilians-Universität, Munich, Germany. [6]Graduate School of Systemic Neurosciences, LMU Munich, Munich, Germany. [7]Department of Radiology, LMU University Hospital, LMU Munich, Munich, Germany. [8]Biomedical Center (BMC), Biochemistry, Faculty of Medicine, LMU Munich, Munich, Germany. [9]Munich Cluster for Systems Neurology (Synergy), Munich, Germany. [10]Brain and Mind Centre, Medical Imaging Sciences, Faculty of Medicine and Health, The University of Sydney, Sydney, NSW, Australia. [11]Australian Nuclear Science and Technology Organisation (ANSTO), Sydney, NSW, Australia. [12]Ageing Epidemiology Research Unit (AGE), School of Public Health, Imperial College London, London, UK. [13]Division of Neuroscience, University of Sheffield, Sheffield, UK. [14]Department of Neurology, LMU University Hospital, LMU Munich, Munich, Germany. [15]German Cancer Consortium (DKTK), Munich, Germany. [16]These authors contributed equally: Jochen Herms, Lars Paeger. ✉e-mail: lars.paeger@dzne.de

