## [Transparent Peer Review file · Nature Communications]

Early Locus Coeruleus noradrenergic axon loss drives olfactory dysfunction in Alzheimer's disease

Corresponding Author: Dr Lars Paeger

Version 0:

Reviewer comments:

Reviewer #1

(Remarks to the Author)

In this study, Meyer et. al. investigated the mechanism underlying hyposmia, one of the early non-cognitive symptoms of Alzheimer's disease (AD), using the APPNL-G-F mouse model of amyloid pathology, postmortem human brain tissues and data from AD patients. To this end, they focused on the role of noradrenergic locus coeruleus (LC) dysfunctions in hyposmia, since the neurotransmitter noradrenalin (NA) is known to modulate olfactory information processing, and LC is one of the earliest brain regions affected in AD. They found that noradrenergic inputs and NA release in the olfactory bulb (OB) significantly decreased, and olfaction was impaired in the APPNL-G-F mice. Interestingly, these changes were observed at the age of 2 months before pronounced appearance of extracellular amyloid plaques. They found that the number of microglia was increased in the OB, whereas transcriptional analysis did not find evidence for "disease-associated" microglial response in the OB. As a mechanism of loss of noradrenergic axons in the OB, they found that OB microglia from the APPNL-G-F mice showed increased phagocytic activity of synaptosomes in vitro and noradrenergic fibers in the brains. Noradrenergic inputs to the OB were more decorated with "find-me"- and "eat-me"-signals including phosphatidylserine and MFG-E8 in the APPNL-G-F mice than in the control mice. This neuronal damage seemed to be intrinsically caused by expression of APPNL-G-F protein in the LC neurons. They also showed that knockout of translocator protein 18 kDa (TSPO), which is critical for microglial activity, reduced phagocytosis of noradrenergic fibers and prevented olfactory deficits in the APPNL-G-F mice. Finally, they showed that patients with prodromal AD displayed elevated TSPO-PET signals in the OB, similarly to the APPNL-G-F mice and early LC axon degeneration in post-mortem OBs in patients with early AD.

Overall, this is a very interesting study with extensive data and analyses. However, there are several concerns with the interpretation of the data. Also, the relevance of the proposed mechanism of LC degeneration caused by APPNL-G-F overexpression in the LC of AD mice to the LC loss and hyposmia in AD patients is not clear.

Specific comments

- 1) In Fig. 1, authors showed that the NET fiber density decreased, and the number of microglia increased at the age of 2 months before pronounced amyloid plaque formation in APPNL-G-F mice. Did authors analyze fiber density other than NET+ fibers? Also, what drove microglial increase in APPNL-G-F mouse brains?
- 2) In Fig. 3f, why microglial cells isolated from APPNL-G-F mouse brains showed increased phagocytic activity of synaptosome compared to microglial cells from C57BL/6J control mouse brains? Since authors hypothesized enhanced phagocytic activity in APPNL-G-F mouse brains from transcriptome analysis, it would be helpful to show or discuss specific mechanisms.
- 3) In Fig. 4h and i, authors showed that there was no difference in the volume of contact points between microglia and LC axons between control and APPNL-G-F mice (Fig. 4h,i). Does this mean that microglia in APPNL-G-F mouse brains only engulfed damaged axons as a protective response but did not engulf intact axons pathologically?
- 4) Related to Fig. 4e and i, how did TSPO knockout specifically reduce phagocytic activity of microglia without altering volume of microglia and volume of contact points between microglia and LC fibers?
- 5) Related to Fig. 5, how did overexpression of APPNL-G-F protein damage LC fibers? Unlike APP knockin, overexpression of APP causes axonal transport defects and neuronal dysfunctions, which may be different from neurodegeneration in AD.

Please carefully discuss this point.

6) Related to Fig. 6a, did authors analyzed tau pathology, synuclein pathology and neuronal loss in the LC regions in prodromal AD and AD patients? Also, one of prodromal AD patients (#13) do not show amyloid pathology and is not likely to be AD.

7) Related to Fig. 6a-f, NET fiber density, TSPO-PET and score for order identification task do not correlate well in prodromal AD.

8) Related to Fig. 6g-j, it is surprising that TSPO-PET signal is detected in the OB, but not in the cortex where extensive amyloid plaques and microglial activation are observed. Could author comment on this point?

9) In Extended Fig.1, data from 12-month-old mice are missing.

(Remarks on code availability)

Reviewer #2

(Remarks to the Author)

The expertise I have relevant to this paper is TSPO imaging, and hence I was asked to review just that section of the manuscript.

Nevertheless, I thought that overall this was a very interesting paper on an important topic and its results are of significance to the field and ultimately to patients.

Regarding the TSPO imaging, the authors in my view have used TSPO imaging appropriately, by using it to ask whether there are differences in microglial density between the three groups (rather than phenotype).

I have a few comments.

The authors used post mortem brain tissue to show that there was axon loss in AD and pAD relative to control brains. They then wanted to determine if this axon loss was associated with an increase in microglial number, which they addressed with PET imaging in patients/controls. The result is they show the axon loss in post mortem tissue and the increase in microglial number in living people. There may be a reason, but it is not immediately obvious to me why they didn't look for an increase in microglial number in those same post mortem brains where they demonstrated axon loss (and potentially then correlating axon loss with microglial number within sample). The paper would certainly be strengthened by doing so.

I would also query the claim that controls are age matched. Unless I have misunderstood, the median ages for control, pAD and AD are 64, 77, 87 years respectively. Almost all of the controls are age 65 and below, whereas almost all of the AD/pAD are 73 and above. This is important because axon loss is partly a function of age. The differences the authors report between control and AD/pAD may well be greater than what would be expected with age, but the current dataset leaves this as an uncertainty. I know post mortem sample availability is always an issue, but I would strongly suggest the authors find a few more samples to mitigate the current age differences between the groups to give us confidence that the results are not explained by age differences.

Similarly I think the finding would be strengthened by showing that the axon loss is specific to NET+ fibres and that other fibres are unaffected (or much less affected).

A very minor point is that the control brains in figures 6a,b,c are referred to as coming from healthy controls (line 294) which is an unusual way to describe someone who is dead.

(Remarks on code availability)

Reviewer #3

(Remarks to the Author)

This is a novel and timely study showing that early olfactory deficits in an APP knock in mouse model are driven by the selective degeneration of noradrenergic axons from the locus coeruleus to the olfactory bulb, mediated by microglial clearance through phosphatidylserine and MFG-E8 signalling, already before the formation of plaques. Core findings are nicely validated in human prodromal AD patients, which further increases the impact of the study.

I have only few comments related to the two-photon imaging:

Figure 2:

- How plots were generated is a bit unclear with regard to the averaging. Is the fluorescence averaged across the whole frame? And then across trials or sessions?
- It looks like from the heat map that some cells / FOVs / mice are inhibited rather than being excited. This is masked in the 2f, the grand average. I would like to see a quantification of the number of cells / sessions / mice / FOVs significantly excited / inhibited. This could give more insight into the mechanism (i.e. NA has a greater inhibitory effect on the circuit in NLGFs). If it's averaged across the whole FOV, is it possible that some regions are excited and inhibited? Again this could mask the effect in the averaging.
- Did the authors ever do imaging at 6 months? It would be interesting to see the effect of NA when there is significant plaque burden but I don't see that as a requirement for a revision.

Figure 3d:

I don't think this correlation is real (https://en.wikipedia.org/wiki/Simpson's_paradox). If it is the authors need to improve their explanation of the regression. This is not central to their results though.

Marc Aurel Busche

(Remarks on code availability)

Reviewer #4

(Remarks to the Author)

This study by Meyer et al. investigates the connection between early olfactory dysfunction in Alzheimer's Disease (AD) and the loss of noradrenergic (NA) axons originating from the locus coeruleus (LC) in a mouse model of AD, as well as in human postmortem tissue. The authors combine behavioral testing, advanced imaging techniques (TSPO-PET and two-photon microscopy), and RNA sequencing to explore mechanisms underlying early olfactory deficits. Their findings suggest that microglia-mediated clearance of LC axons in the olfactory bulb (OB) plays a key role in the pathogenesis, highlighting the potential of olfactory dysfunction as an early event leading to memory loss in AD.

The study addresses an important but underexplored area in AD research, providing mechanistic insights into olfactory dysfunction, which is a well-documented yet poorly understood early symptom of AD. While the work offers many interesting findings on OB dysfunction, the basic deficit in odor processing associated with early AD remains unclear.

Specifically, although the authors demonstrate impaired olfaction at the behavioral level from 3 months of age, the precise mechanisms by which familial AD mutations impair olfactory coding remain unresolved. It is unclear what specific aspects of olfaction are affected by lower NA levels and increased phagocytosis by microglia. The authors assess odor-induced NA release using two-photon microscopy with a single odor at a single concentration. The important control with mutated GRAB sensor insensitive to NA is not provided.

Furthermore, this experiment does not provide insight into how odor detection and discrimination are affected in the OB of AD model mice. No quantitative analysis is presented regarding the amplitude of the neural response, trial-to-trial variability, adaptability, or the fraction of neurons responding to a given odor. Additionally, there is no analysis of whether the neural responses convey information about the identity of the presented odors and decoding accuracy.

It is unclear whether the deficits in olfaction arise from the OB specifically, or whether other regions of the olfactory system, such as the piriform cortex, also contribute to the sensory deficits in young AD mice. This uncertainty is further compounded by the lack of electrophysiological phenotype of mitral cell activity in the OB and by the lack of olfaction recovery following chemogenetic activation of LC neurons.

(Remarks on code availability)

Version 1:

Reviewer comments:

Reviewer #1

(Remarks to the Author)

Authors addressed this reviewer's concerns on data from mouse models, sufficiently.

As for data from human study, I still have concerns since authors did not provide additional evidence. Instead, authors claimed that they did not have samples to perform required experiments or enough data points to reach statistical

significance.

(Remarks on code availability)

Reviewer #2

(Remarks to the Author)

All my comments have been addressed satisfactorily. I commend the authors for their work

David Owen

(Remarks on code availability)

Reviewer #3

(Remarks to the Author)

The authors have fully addressed my concerns. The study is innovative, translational, and ready for publication; I recommend acceptance.

(Remarks on code availability)

Reviewer #4

(Remarks to the Author)

The authors have strengthened their conclusions in the revised manuscript. In my view, it is now ready for publication.

(Remarks on code availability)

To the reviewers:

We would like to thank all reviewers for the time and effort investing in the review of our submitted manuscript to *Nature Communications*. We feel the justified comments and critical questions have improved the quality of the manuscript significantly. We hope to have addressed all relevant points to the satisfaction of the reviewers and provide detailed responses point-by-point referring to the relevant changes in the manuscript.

REVIEWER COMMENTS

Reviewer #1 (Remarks to the Author):

In this study, Meyer et. al. investigated the mechanism underlying hyposmia, one of the early non-cognitive symptoms of Alzheimer's disease (AD), using the APPNL-G-F mouse model of amyloid pathology, postmortem human brain tissues and data from AD patients. To this end, they focused on the role of noradrenergic locus coeruleus (LC) dysfunctions in hyposmia, since the neurotransmitter noradrenalin (NA) is known to modulate olfactory information processing, and LC is one of the earliest brain regions affected in AD. They found that noradrenergic inputs and NA release in the olfactory bulb (OB) significantly decreased, and olfaction was impaired in the APPNL-G-F mice. Interestingly, these changes were observed at the age of 2 months before pronounced appearance of extracellular amyloid plaques. They found that the number of microglia was increased in the OB, whereas transcriptional analysis did not find evidence for "disease-associated" microglial response in the OB. As a mechanism of loss of noradrenergic axons in the OB, they found that OB microglia from the APPNL-G-F mice showed increased phagocytic activity of synaptosomes in vitro and noradrenergic fibers in the brains. Noradrenergic inputs to the OB were more decorated with "find-me"- and "eat-me"-signals including phosphatidylserine and MFG-E8 in the APPNL-

G-F mice than in the control mice. This neuronal damage seemed to be intrinsically caused by expression of APPNL-G-F protein in the LC neurons. They also showed that knockout of translocator protein 18 kDa (TSPO), which is critical for microglial activity, reduced phagocytosis of noradrenergic fibers and prevented olfactory deficits in the APPNL-G-F mice. Finally, they showed that patients with prodromal AD displayed elevated TSPO-PET signals in the OB, similarly to the APPNL-G-F mice and early LC axon degeneration in post-mortem OBs in patients with early AD.

Overall, this is a very interesting study with extensive data and analyses. However, there are several concerns with the interpretation of the data. Also, the relevance of the proposed mechanism of LC degeneration caused by APPNL-G-F overexpression in the LC of AD mice to the LC loss and hyposmia in AD patients is not clear.

Specific comments

1) In Fig. 1, authors showed that the NET fiber density decreased, and the number of microglia increased at the age of 2 months before pronounced amyloid plaque formation in APPNL-G-F mice. Did authors analyze fiber density other than NET+ fibers?

We thank the reviewer for this critical comment. We fully agree to the importance of investigating other fiber densities, since other subcortical modulatory systems are known to be affected in the progression of AD as well¹. We have thus performed immunohistochemical analysis of cholineacetyltransferase positive (ChAT⁺) and serotonergic (SERT⁺) fibers in the olfactory bulb of C57BL/6J and APP^{NL-G-F} animals. We show that in contrast to NET⁺ axons, neither degeneration of ChAT⁺ neurites nor SERT⁺ axons at 3 months of age can be detected in APP^{NL-G-F} mice. We hence conclude

that the specific loss of noradrenergic LC axons reflects the vulnerability of OB-projecting LC neurons. These data are added to Extended Data Fig. 1 and are furthermore mentioned in the “Results” and “Discussion”

Also, what drove microglial increase in APPNL-G-F mouse brains?

At this point we can only speculate about the underlying mechanism for an increased microglial density. Our data indicates an increase in microglial cell numbers starting at an age of two months onwards. We cannot detect the concomitant deposition of extracellular β -amyloid ($A\beta$) plaques. However, an increase in extracellular $A\beta$ -oligomers has been described to precede the formation of $A\beta$ -plaques in APP^{NL-G-F} mice. At early stages, microglia may already increase phagocytic activity and prevent plaque formation². In this context, a recent paper also shows that microglia are spatially more associated to $A\beta$ -oligomers than to plaques in APP^{NL-G-F} mice³. Earlier work has illustrated an increased number of microglia in response to elevated $A\beta$ levels. Thus, $A\beta$ -oligomers as a trigger for microglial proliferation and increased cell number constitutes one possible explanation⁴. Additionally, recent work illustrates a stark modulation of microglia function by the neuromodulator NA⁵⁻⁹. That is, the loss of NA innervation and consequently NA release in the olfactory bulb might indeed exert further effects on microglial function¹⁰. As the effects of NA, however, might be strongly state- and context-dependent and additionally differ between anatomical locations, it is impossible to delineate the exact contribution of a decreased NA signaling on microglia in the OB. We completely agree that these are fundamental questions and would like to emphasize that we already started to outline specific projects in the future addressing these critical points specifically.

2) In Fig. 3f, why microglial cells isolated from APPNL-G-F mouse brains showed increased phagocytic activity of synaptosome compared to microglial cells from C57BL/6J control mouse brains? Since authors hypothesized enhanced phagocytic activity in APPNL-

G-F mouse brains from transcriptome analysis, it would be helpful to show or discuss specific mechanisms.

We thank the reviewer for the comment, and take the opportunity to clarify why we hypothesize an increased phagocytic activity in APP^{NL-G-F} mice. With respect to our transcriptomic data, we did not find a significant change in modules consisting of genes annotated with the GO-term “Phagocytosis”. However, we found several gene modules significantly upregulated that were annotated to the GO-term “Synapse” (we have added heatmaps of the genes with GO “Phagocytosis” and “Synapse” as additional illustrations to Extended Data Fig. 7). Genes annotated with the GO-term “Synapse” included several genes that have been shown to play a role in synapse remodeling and synaptic restructuring and may thus include genes for synaptic pruning. A number of genes is suggested to regulate phagocytosis directly or indirectly including but not limited to *Chma7*, *Lrrtm4*, *Bln2*, *Epha4*. Based on this, we suggested that microglia cells might show elevated phagocytic activity even in the absence of upregulated genes annotated with “Phagocytosis” or being counted as genes DE in the DAM-response.

Furthermore, decreased NA release might contribute to altered phagocytic activity and gliosis significantly^{6,10,11}. NA is a critical modulator of microglia activity and may have differential effects as mentioned in the discussion.

3) In Fig. 4h and i, authors showed that there was no difference in the volume of contact points between microglia and LC axons between control and APPNL-G-F mice (Fig. 4h,i). Does this mean that microglia in APPNL-G-F mouse brains only engulfed damaged axons as a protective response but did not engulf intact axons pathologically?

This is indeed an important question. We sought to address this in our discussion. It may be the case that this was not particularly clear enough. We hypothesize that at that particular stage without the presence of overt A β -plaque deposition, microglia do not necessarily form enhanced contact

points to other cell types. However, the presence of the “eat-me”-signal PS and MFG-E8 instigate microglia to phagocytose LC axons. That same signaling mechanism is used during adult neurogenesis and synaptic pruning in the OB, rendering this area sensitive to this signal. We hypothesize that microglia engulf axons in their function to surveil tissue integrity. Thus, unaltered contact points still may result in an elevated phagocytosis due to the increased probability to detect “eat-me” signals on LC axons. Furthermore, we do notice a slight trend towards a higher number of contact points in APP^{NL-G-F} and APP^{NL-G-F} x TSPO-KO microglia, without reaching statistical significance (Fig. 4i).

4) Related to Fig. 4e and i, how did TSPO knockout specifically reduce phagocytic activity of microglia without altering volume of microglia and volume of contact points between microglia and LC fibers?

The reviewer is indeed right that reduced phagocytosis can as well correlate to an anatomical change in microglia cells. Recently, Fairley and colleagues have assessed mitochondrial metabolism in microglia with respect to TSPO function¹². Specifically, TSPO-KO was shown to result in decreased phagocytosis due to a shift from oxidative phosphorylation towards glycolysis¹². In their work, they describe glycolytic activity to be insufficient to provide necessary amounts of ATP for phagocytosis. In a series of elegant *in vitro* experiments, they could show that TSPO-KO leads to the recruitment of a hexokinase promoting the shift towards less efficient glycolysis (we have specified the rationale for investigating the effect of TSPO-KO now in the relevant part of the “Results”). We leveraged this in order to modulate phagocytic activity by genetic means to prove the phagocytosis of LC axons by microglia and finally the rescue of olfaction in APP^{NL-G-F} x TSPO-KO mice. In addition, our own group has shown that TSPO-KO leads to a slightly increased morphological complexity of microglia in the cortex¹³. In Fig. 4e a slight trend towards an increased microglia volume is noticed which is the reason for volume normalization in panel on the right. Concomitantly, as referred to in the response to the

reviewer's "specific comment 3)" there is a slight trend towards a higher number of contact points in APP^{NL-G-F} and APP^{NL-G-F} x TSPO-KO microglia, without reaching statistical significance.

5) Related to Fig. 5, how did overexpression of APP^{NL-G-F} protein damage LC fibers? Unlike APP knockin, overexpression of APP causes axonal transport defects and neuronal dysfunctions, which may be different from neurodegeneration in AD. Please carefully discuss this point.

We thank the reviewer for addressing this specific and important issue. Indeed, we can assume that overexpression of mutant APP via AAVs might act different on LC neurons compared to the mechanism in the APP^{NL-G-F} transgenic mice, where the transgene is expressed at physiological levels. The overexpression of the mutant APP, which is more prone to be processed into toxic A β 42 peptides, might indeed be even more toxic. We would like to point out that the rationale for this experiment was to induce the production of A β 42 peptides only in LC neurons. We could not think about an alternative approach to do so. To account for the short discussion, we have now added a substantial part in the manuscript referring to recent work, which might add valuable insights and should foster future research depicting the selective vulnerability of LC axons in the APP context. The reviewer is indeed right, that overexpression of human APP in LC neurons might lead to axonal transport deficits, however, this has indeed been also attributed to the Swedish mutation, which is also a part of this mouse model. That is, axonal transport deficits might be indeed a general mechanism that can be applied to both conditions, though certainly to different extents. In support of that, recent work illustrates a mitochondrial transport deficiency in APP expressing neurons of transgenic mice¹⁴. Since LC axons are long, thin and unmyelinated it is reasonable to speculate that any transport deficits might especially affect axons that project to regions several millimeters distant from their originating soma. We have added a relevant part in "Discussion". We feel that this will clearly stimulate novel

and sophisticated research projects. *In vivo* imaging of axonal transport is now possible and we think these are very important details and subjects for follow-up studies, especially when it comes to the pharmacotherapy of neurodegenerative diseases.

6) Related to Fig. 6a, did authors analyzed tau pathology, synuclein pathology and neuronal loss in the LC regions in prodromal AD and AD patients? Also, one of prodromal AD patients (#13) do not show amyloid pathology and is not likely to be AD.

We do agree with the author, that in the light of the emerging importance of co-pathologies in neurodegenerative diseases this is indeed an important question. We thus have added further stainings to depict the presence of tau and α -synuclein (α Syn) pathology. Unfortunately, with the tissue available we cannot quantify the absolute levels of pathology and cell numbers. However, we found that all prodromal AD and AD patients showed tau-pathology and only 2 out of 15 patients among the unaffected healthy brain donors and AD brain donors showed α Syn co-pathology. We added this to the new Extended Data Fig. 11. We otherwise agree with the note on patient (#13) and have excluded the data obtained from this particular donor and updated the figure and statistics. This has not led to any changes in the conclusion but was an importantly raised point and significantly enhanced the quality of the data set. We thank the Reviewer for his thorough examination of the pathology.

7) Related to Fig. 6a-f, NET fiber density, TSPO-PET and score for odor identification task do not correlate well in prodromal AD.

We thank the Reviewer for the note. Indeed, TSPO-PET and odor identification scores do not correlate well. We would like to point out, that the study cohort here is particularly small, especially when it comes to the odor identification task. As addressed in our discussion, now several papers clearly depict olfactory dysfunction as a predictor of cognitive

decline and an additional diagnostic tool. Typically, however, these studies include several hundreds of subjects compared to our 47 patients.

8) Related to Fig. 6g-j, it is surprising that TSPO-PET signal is detected in the OB, but not in the cortex where extensive amyloid plaques and microglial activation are observed. Could author comment on this point?

We thank the Reviewer for pointing this out. We focused specifically on an early timepoint in our study. In conjunction with our transcriptomic data, which did not reveal differentially expressed TSPO in microglia of OB, we assume that indeed the increase in TSPO is due to the increased density of microglia cells in the OB. We performed a new set of immunostainings and reconstructed Iba1⁺ and TSPO⁺ volumes in the OB (now included in Extended Data Fig. 12). Indeed, we did not detect a significant intrinsic upregulation of microglial TSPO in the OB, which leads to the conclusion that similarly to the human TSPO-PET, increased signal reflects elevated microglial density rather than an increased TSPO expression on the single cell level and activated microglia. We further would like to refer to our own study, where longitudinal PET imaging of amyloidosis and TSPO revealed first statistical significances at the age of 5 months in the cortex and hippocampus of APP^{NL-G-F} mice¹⁵.

9) In Extended Fig.1, data from 12-month-old mice are missing.

We thank the reviewer for the comment, but would like to point out, that we have indeed never analyzed mice at the age of 12 months. The study's focus was particularly an early phenotype, thus all immunohistochemical work is performed up to an age of six months.

Reviewer #2 (Remarks to the Author):

The expertise I have relevant to this paper is TSPO imaging, and hence I was asked to review just that section of the manuscript.

Nevertheless, I thought that overall this was a very interesting paper on an important topic and its results are of significance to the field and ultimately to patients.

Regarding the TSPO imaging, the authors in my view have used TSPO imaging appropriately, by using it to ask whether there are differences in microglial density between the three groups (rather than phenotype). I have a few comments.

We would like to thank Reviewer #2 for the critical reading and an encouraging comment on the importance of our study. We do agree that in this case we used TSPO-PET to assess the density of microglia in the OB rather than an elevated TSPO-expression phenotype, of which the presence in humans is currently controversially discussed. We would like to further refer to our additional data set, that depicts that also in APP^{NL-G-F} mice elevated TSPO-PET signal clearly reflects an increase in microglia density, as we have evaluated TSPO expression in single microglia cells from the OB of these animals compared to WT. We did not find an elevated TSPO expression on the single microglial level (Extended Data Fig. 12).

The authors used post mortem brain tissue to show that there was axon loss in AD and pAD relative to control brains. They then wanted to determine if this axon loss was associated with an increase in microglial number, which they addressed with PET imaging in patients/controls. The result is they show the axon loss in post mortem tissue and the increase in microglial number in living people. There may be a reason, but it is not immediately obvious to me why

they didn't look for an increase in microglial number in those same post mortem brains where they demonstrated axon loss (and potentially then correlating axon loss with microglial number within sample). The paper would certainly be strengthened by doing so.

We completely agree with the note of the Reviewer. Unfortunately, there are several issues with the human olfactory bulb tissue. First, the bulb is not always collected when a brain is donated to our brain bank but only in the most recent years. Second, and most importantly, a brain with early AD pathology is very rare – about a hand full out of the 200 AD cases in the Munich Brain bank. Third, one olfactory bulb is routinely stored in -80°C and the other one is completely embedded in Paraffin. However, to analyze microglia in human tissue, for example in the cortex, we need thick formalin fixed tissue sections which allow us to reconstruct the entire domain of microglia cells. However, we do not have any olfactory bulb left in formalin and the paraffin tissue cannot be sectioned into 50µm thick sections. Therefore – even though it sounds easy, it is difficult to analyze glia cells in paraffin embedded tissue because the domains are too big relative to the thickness of the sections. That is, we would like to refer to the following publications, which we now have addressed in our “Discussion”. These papers specifically illustrate an increased number of microglia cells in olfactory bulb from AD patients. However, there is no data available on brains with early AD pathology^{16,17}.

I would also query the claim that controls are age matched. Unless I have misunderstood, the median ages for control, pAD and AD are 64, 77, 87 years respectively. Almost all of the controls are age 65 and below, whereas almost all of the AD/pAD are 73 and above. This is important because axon loss is partly a function of age. The differences the authors report between control and AD/pAD may well be greater than what would be expected with age, but the current dataset leaves this as an uncertainty. I know post mortem sample

availability is always an issue, but I would strongly suggest the authors find a few more samples to mitigate the current age differences between the groups to give us confidence that the results are not explained by age differences.

We were able to add tissue from two unaffected controls and a prodromal AD brain donor. The data has been updated and for an overview the new average ages are:

Unaffected controls: n = 9, average age: 71

pAD: n = 8, average age: 77

AD: n = 6, average age: 82

Similarly I think the finding would be strengthened by showing that the axon loss is specific to NET+ fibres and that other fibres are unaffected (or much less affected).

This again is indeed a very important point. With reference to Reviewer's #1 specific comment 1), we have addressed this point in tissue from APP^{NL-G-F} mice. At an early stage, neither degeneration of cholinergic neurites (ChAT⁺) nor serotonergic axons (SERT⁺) can be detected in APP^{NL-G-F} mice. These data (Extended Data Fig.1) are furthermore mentioned in the "Results" and "Discussion".

A very minor point is that the control brains in figures 6a,b,c are referred to as coming from healthy controls (line 294) which is an unusual way to describe someone who is dead.

We thank the Reviewer for referring to this important point. We have changed the respective description to "unaffected control brain donors" and changed also further similar mistakes in the manuscript.

Reviewer #3 (Remarks to the Author):

This is a novel and timely study showing that early olfactory deficits in an APP knock in mouse model are driven by the selective degeneration of noradrenergic axons from the locus coeruleus to the olfactory bulb, mediated by microglial clearance through phosphatidylserine and MFG-E8 signalling, already before the formation of plaques. Core findings are nicely validated in human prodromal AD patients, which further increases the impact of the study.

I have only few comments related to the two-photon imaging:

Figure 2:

- How plots were generated is a bit unclear with regard to the averaging. Is the fluorescence averaged across the whole frame? And then across trials or sessions?

We thank Marc Aurel Busche for addressing the point. We apologize if this was unclear. Within each field of view (FOV), we identified three regions of interest (ROIs), which showed highest baseline fluorescence as a proxy for good GRAB_{NE} expression¹⁸ and to avoid analysis of image regions containing vasculature, which are devoid of GRAB_{NE} and can show fluctuations in diameter during an applied stimulus. We have added a more thorough description of the data acquisition and analysis process to "Methods". These include also the details on the extensive re-performed analysis and additional presentation of the data as requested by the Reviewer. We also refer the reviewer to the related responses in the next point.

- It looks like from the heat map that some cells / FOVs / mice are inhibited rather than being excited. This is masked in the 2f, the grand average. I would like to see a quantification of the number of cells / sessions / mice / FOVs significantly excited / inhibited. This could give more insight into the mechanism (i.e. NA has a greater inhibitory effect on the circuit in NLGFs). If it's averaged across the whole FOV, is it possible that some regions are excited and inhibited?

Again this could mask the effect in the averaging.

We agree that the illustration in the grand average only is not very informative. We have thus reassessed our imaging data and have binned the whole imaging frames in 6x6 different ROIs, yielding a total of 36 ROIs per FOV. The 3 brightest of these ROIs were previously employed for analysis as described in the response above. Now for the updated analysis, all 36 ROIs from all 3 FOVs from three animals per genotype, alongside the GRAB_{NE} sensor control (as reasonably requested from Reviewer #4) are now illustrated in Fig. 2 (e-h). The total 324 ROIs per group have been sorted according to its specific response upon odor (vanilla) stimulation from increase in dF/F to decreases within the chosen analysis window of frame 86-91 (see also extended Fig. 3c). With respect to the Reviewers comment we however have to clearly point out, that we cannot categorize the response into excitatory or inhibitory. The fluorescence is solely a function of binding the neurotransmitter NA, which otherwise binds to a variety of different receptor subtypes exerting either excitatory or inhibitory effects¹⁹. However, to account for the request, we have determined which of the 324 ROIs for WT and APP^{NL-G-F} mice respond significantly with increased or decreased fluorescence (Fig. 2i) and found that the majority of ROIs in WT animals respond with an increase in fluorescence (or NA release). This relationship is significantly altered in APP^{NL-G-F} mice. Neuronal responses are more complex, as cells in the OB likely express numerous subtypes and levels of noradrenergic receptors that differ in their

affinity to the endogenous ligand, yielding divergent effects on downstream neuronal excitability. This intricate relationship has been addressed by recent elegant work from Geng and colleagues. Here, optogenetically triggered NA release differentially affects mitral cell activity, with approximately one-third of the population showing excitation, another third inhibition, while the last third remains unresponsive¹⁹.

Consequently, the neuronal response is a function of the released NA concentration and might differ markedly. Without precisely controlling for NA release by optogenetics and determining NA release quantitatively by 2P-imaging in conjunction with 2P Ca²⁺ imaging, no conclusive data can be expected here. With respect to this, we would also like to refer to the response to Reviewer's 4 question on the neural response, which we sought to address by applying NA while performing acute brain slice recordings from mitral cells of WT and APP^{NL-G-F} mice (Extended Data Fig. 4).

Nevertheless, in our opinion the refined and differentiated analysis requested by Marc Aurel Busche clearly improves the quality of the manuscript and provides important insights for future experiments. We sought to address these points in "Discussion".

- Did the authors ever do imaging at 6 months? It would be interesting to see the effect of NA when there is significant plaque burden but I don't see that as a requirement for a revision.

We thank the reviewer for that interesting thought and we indeed agree, that this would be an interesting aspect, which we are planning to address in follow-up studies.

Figure 3d:

I don't think this correlation is real (https://en.wikipedia.org/wiki/Simpson's_paradox). If it is the authors

need to improve their explanation of the regression. This is not central to their results though.

We thank Marc Aurel Busche for pointing this out and we agree that the illustration as presented in our initial submission can lead to this conclusion. We gratefully take the opportunity to clarify this point. In Figure 3d, we report a direct comparison between the fold-changes observed in 8-month-old cortex and 2-month-old OB (using the same mouse model and MG isolation methods). Simpson's paradox typically emerges when a correlation is confounded by not respecting obvious grouping factors that would tilt the correlation into the other direction. We repeated this analysis with the most intuitive grouping factors, in this case the annotation of MG-associated DAM and homeostatic genes. While there was a slight positive Pearson's r coefficient for the damage-associated MG genes, this correlation was very weak and not statistically significant ($R = 0.066$, $p = 0.48$). In fact, most of the $\sim 3,100$ genes that both datasets had in common are not annotated with a DAM or homeostatic signature ("NA"), thus, the strong negative correlation is driven by these genes. While the function of most of these genes is unknown, we still anticipate their expression levels to represent a biological state that is unrelated to the changes observed in an older mouse cortex. We have re-phrased our statement regarding this association by better describing our regression model in the "Results" and added the regression below to Extended Data Fig. 6.

Marc Aurel Busche

Reviewer #4 (Remarks to the Author):

This study by Meyer et al. investigates the connection between early olfactory dysfunction in Alzheimer's Disease (AD) and the loss of noradrenergic (NA) axons originating from the locus coeruleus (LC) in a mouse model of AD, as well as in human postmortem tissue. The authors combine behavioral testing, advanced imaging techniques (TSPO-PET and two-photon microscopy), and RNA sequencing to explore mechanisms underlying early olfactory deficits. Their findings suggest that microglia-mediated clearance of LC axons in the olfactory bulb (OB) plays a key role

in the pathogenesis, highlighting the potential of olfactory dysfunction as an early event leading to memory loss in AD.

The study addresses an important but underexplored area in AD research, providing mechanistic insights into olfactory dysfunction, which is a well-documented yet poorly understood early symptom of AD. While the work offers many interesting findings on OB dysfunction, the basic deficit in odor processing associated with early AD remains unclear.

Specifically, although the authors demonstrate impaired olfaction at the behavioral level from 3 months of age, the precise mechanisms by which familial AD mutations impair olfactory coding remain unresolved. It is unclear what specific aspects of olfaction are affected by lower NA levels and increased phagocytosis by microglia. The authors assess odor-induced NA release using two-photon microscopy with a single odor at a single concentration. The important control with mutated GRAB sensor insensitive to NA is not provided.

We thank the reviewer for specifically pointing out the missing control with the mutated GRAB to substantiate the sensitivity and suitability of the GRAB_{NE} biosensor in this context. We have thus added a data set of WT animals (n = 3) that have been injected with the mutant variant of the GRAB sensor as published by Feng and colleagues²⁰. We added these data to Fig. 2(f-h). Furthermore, the Reviewer is certainly right that using a single odor and concentration might limit the conclusiveness on the observed neuromodulatory phenotype. Upon ethical approval, we have thus performed additional surgeries in WT and APP^{NL-G-F} mice at the age of 2 months and performed NA release imaging with two further chemically defined odors after a recover period of one month (in addition to Lemon, for which we already acquired data in the first cohort of animals). Each odorant elicited a decreased NA release response in APP^{NL-G-F} mice compared to WT controls (Fig. 2j). Collectively, these data have strengthened the conclusiveness as in summary of all odorants an even

more robust phenotype has been observed. We further aimed at comparing two different concentrations of the two new odors, of which the response picture was less clear in trials with lower concentrations than the responses to higher concentrations (Extended Data Fig. 3d).

Furthermore, this experiment does not provide insight into how odor detection and discrimination are affected in the OB of AD model mice. No quantitative analysis is presented regarding the amplitude of the neural response, trial-to-trial variability, adaptability, or the fraction of neurons responding to a given odor. Additionally, there is no analysis of whether the neural responses convey information about the identity of the presented odors and decoding accuracy.

We agree with the reviewer's comment, that indeed our experiment is not revealing the detailed information at which part olfaction is affected specifically. We agree that these are indeed interesting and important questions that arise from the data we present here. However, we would like to respectfully point out that we think that these questions are out of scope of the current manuscript, since it was our goal to determine whether an early degeneration of the LC-NA system is affecting olfaction in an β -amyloid mouse model of AD. We do present, also with the help of Reviewer's #4 comments, strongly supporting data for this hypothesis. The exact downstream mechanisms of impaired NA release on multiple levels of olfactory decoding, have yet to be determined. However, we would like to specifically point out that such work is typically subject to stand-alone publications in the olfactory field and especially in this case, cannot be addressed by imaging of the neural response²¹⁻²⁶ (please also see our reply to Marc Aurel Busche's comments).

We would like to refer to recently published work in ***Nature Communications*** by Geng and colleagues¹⁹, who have addressed the contribution of NA to olfaction in WT animals specifically. Their elegant analysis of the noradrenergic modulation of olfaction, fills a complete paper

with in total eight major figures. While this reflects the complexity of the system, it also illustrates that unmasking the downstream effects of altered NA signaling exceeds the scope of our submitted manuscript. Yet, their work also provides critical proof of principle experiments that support the findings in our comparison of WT and APP^{NL-G-F} mice. Precisely, Geng and colleagues assessed chemogenetic inhibition of NA release in combination with the buried food task. Indeed, they find that decreased NA release drastically increases the time to find the buried food pellet. This is very much in line with our data, depicting that reduced NA by LC-specific axon loss is underlying the olfactory deficits in APP^{NL-G-F} mice as well as in Dbh-Cre animals that express APP^{NL-G-F} specifically in LC neurons.

The same paper also illustrates that an experiment on the amplitude of neural responses will likely lead to data with limited conclusions. Geng and colleagues assess the effect of optogenetically triggered NA release on mitral cell activity and find that response distribute into each one third for excitatory and inhibitory effects, while the last third remains unresponsive. This further illustrates differential adrenergic receptor expression and likely reflects the responses of different mitral cell subtypes. Although the OB bulb is composed of layers harboring spatially segregated mitral and granule cells, subtypes within these layers exist ^{25,27-29}. A variety of different adrenergic receptors (AR) are expressed in the OB ^{30,31}. We do not have the ability to clearly distinguish between different cell types in the olfactory bulb neither in WT animals nor in APP^{NL-G-F} mice. That is, we cannot assign activity of individual neurons to a specific type of cell. Furthermore, a single neuron can even express a variety of different ARs leading to a complex responsive pattern that is critically dependent on NA concentration and the affinities of expressed ARs ³². Furthermore, to quantitatively compare neural responsiveness between WT and APP^{NL-G-F} mice, we would need to identify comparable regions within the olfactory bulb that most prominently receive odor information from the same receptors and glomeruli, respectively. In addition, as already pointed out, we would need to correlate neural responses with increases or decreases

in NA upon odor stimulation and to compare to APP^{NL-G-F} need to control NA release, specifically (optogenetically).

However, we understand that the Reviewer seeks to improve the conclusiveness of our manuscript, and we agree that this is interesting information. That is, we furthermore sought to address the Reviewer's question in an acute slice experiment by measuring the response of individual mitral cells to NA (30 μ m) application. The individual response reflects the diversity presented in Gengs's and colleague's paper. Importantly, we found that mitral cells in APP^{NL-G-F} mice showed a trend towards an overall decreased responsiveness of mitral cells to the application of the same concentration of NA (Extended Data Fig. 4g-j).

In addition, we have used further odorants as stimuli as well as two concentrations of these odorants while assessing NA release in the OB. These data are now presented in Fig. 2 and Extended Data Fig. 3, in conjunction with the new analysis of the responses as requested by Marc Aurel Busche. The decreased NA release is detectable irrespectively of a given odor and clearly identifies the NA deficiency due to the loss of NA axons.

We hope to have convinced the Reviewer that these intricately important questions need to be addressed in future experiments and separate publications as the workload (including obtaining and breeding appropriate mice on a homozygous disease background) and the experiments needed are out of scope of our paper. We sought to address these points in "Discussion".

It is unclear whether the deficits in olfaction arise from the OB specifically, or whether other regions of the olfactory system, such as the piriform cortex, also contribute to the sensory deficits in young AD mice. This uncertainty is further compounded by the lack of

electrophysiological phenotype of mitral cell activity in the OB and by the lack of olfaction recovery following chemogenetic activation of LC neurons.

We agree with the reviewer. We thus have analyzed in detail the piriform cortex as the main cortical area for olfactory input. According to the paper of Geng and colleagues, which have shown strong noradrenergic modulation of the anterior PiCTX, we differentiated the anterior and posterior PiCTX. We have performed extensive immunohistochemical analysis which is shown in Extended Data Fig. 1. We show that indeed reduced fiber density in the posterior PiCTX of APP^{NL-G-F} mice can be detected at an age of three months. However, the anterior PiCTX, subject to substantial NA modulation, retains an unaltered LC axon density. We feel that this data clearly enhances the quality of the manuscript and fosters future research, especially considering multimodal imaging in patients while performing olfactory tests.

With respect to the lack of an intrinsic electrophysiological phenotype, our interpretation was rather that indeed impaired olfaction is more likely to be derived from an impaired NA release, rather than by a change in the responsiveness of mitral cells in first place. As mentioned in the previous response, to substantiate these data, we also added a dataset on the responsiveness of mitral cells to NA in acute brain slices from APP^{NL-G-F} and WT mice (Extended Data Figure 4).

We furthermore repeated the chemogenetic stimulation experiment with a prolonged treatment of Dbh-Cre animals injected with an AAV expressing an excitatory DREADD channel (hM3DGq) animal. In a recent paper, this protocol has led to recurrent increase of NET density in a mouse model of autism³³. To account for the controversially discussed mechanism of action of systemically injected CNO, we chose to treat the animals with Clozapine (Clo; 0,03 mg/kg BW) instead^{34,35}. Injection of Clo resulted in pronounced

and reliable induction of LC neuron activation in DREADD expressing animals as indicated by co-immunostaining against Th, and cfos. Importantly, control animals did not show substantial cfos immune-labelling in LC neurons. We added these data to Extended Data Fig.5 and the “Results”. We could not detect a recovery of olfactory behaviour in the animals treated with Clozapine for a period of five days. In our view, this is clearly indicative for the mentioned structure to function relationship of NA axons and the subsequent NA release in the OB.

References to point-by-point response

1. Ehrenberg, A. J. *et al.* Priorities for research on neuromodulatory subcortical systems in Alzheimer’s disease: Position paper from the NSS PIA of ISTAART. *Alzheimer’s Dementia* (2023) doi:10.1002/alz.12937.
2. Feng, W. *et al.* Microglia prevent beta-amyloid plaque formation in the early stage of an Alzheimer’s disease mouse model with suppression of glymphatic clearance. *Alzheimer’s Res. Ther.* **12**, 125 (2020).
3. Tang, J. *et al.* Associations of amyloid- β oligomers and plaques with neuropathology in the AppNL-G-F mouse. *Brain Commun.* **6**, fcae218 (2024).

4. Fontana, I. C. *et al.* Amyloid- β oligomers in cellular models of Alzheimer's disease. *J. Neurochem.* **155**, 348–369 (2020).
5. Xu, H., Rajsombath, M. M., Weikop, P. & Selkoe, D. J. Enriched environment enhances β -adrenergic signaling to prevent microglia inflammation by amyloid- β . *Embo Mol Med* **10**, (2018).
6. Heneka, M. T. *et al.* Locus ceruleus controls Alzheimer's disease pathology by modulating microglial functions through norepinephrine. *Proceedings Of The National Academy Of Sciences Of The United States Of America* **107**, 6058–6063 (2010).
7. Stowell, R. D. *et al.* Noradrenergic signaling in the wakeful state inhibits microglial surveillance and synaptic plasticity in the mouse visual cortex. *Nature Neuroscience* **22**, 1782–1792 (2019).
8. Gyoneva, S. & Traynelis, S. F. Norepinephrine Modulates the Motility of Resting and Activated Microglia via Different Adrenergic Receptors*. *J Biol Chem* **288**, 15291–15302 (2013).
9. Liu, Y. U. *et al.* Neuronal network activity controls microglial process surveillance in awake mice via norepinephrine signaling. *Nature Neuroscience* **22**, 1771–1781 (2019).
10. Le, L. *et al.* Noradrenergic signaling controls Alzheimer's disease pathology via activation of microglial β 2 adrenergic receptors. *bioRxiv* 2023.12.01.569564 (2023) doi:10.1101/2023.12.01.569564.
11. Heneka, M. T. *et al.* Locus ceruleus degeneration promotes Alzheimer pathogenesis in amyloid precursor protein 23 transgenic mice. *The Journal of neuroscience : the official journal of the Society for Neuroscience* **26**, 1343–1354 (2006).
12. Fairley, L. H. *et al.* Mitochondrial control of microglial phagocytosis by the translocator protein and hexokinase 2 in Alzheimer's disease. *Proc. Natl. Acad. Sci.* **120**, e2209177120 (2023).
13. Shi, Y. *et al.* Long-term diazepam treatment enhances microglial spine engulfment and impairs cognitive performance via the mitochondrial 18 kDa translocator protein (TSPO). *Nat Neurosci* 1–13 (2022) doi:10.1038/s41593-022-01013-9.
14. Vaillant-Beuchot, L. *et al.* The amyloid precursor protein and its derived fragments concomitantly contribute to the alterations of mitochondrial transport machinery in Alzheimer's disease. *Cell Death Dis.* **15**, 367 (2024).

15. Sacher, C. *et al.* Longitudinal PET Monitoring of Amyloidosis and Microglial Activation in a Second-Generation Amyloid- β Mouse Model. *J. Nucl. Med.* **60**, 1787–1793 (2019).
16. Kohl, Z. *et al.* Distinct Pattern of Microgliosis in the Olfactory Bulb of Neurodegenerative Proteinopathies. *Neural Plast.* **2017**, 3851262 (2017).
17. Murray, H. C. *et al.* Lamina-specific immunohistochemical signatures in the olfactory bulb of healthy, Alzheimer's and Parkinson's disease patients. *Commun. Biol.* **5**, 88 (2022).
18. Scekcic-Zahirovic, J. *et al.* Cortical hyperexcitability in mouse models and patients with amyotrophic lateral sclerosis is linked to noradrenaline deficiency. *Sci. Transl. Med.* **16**, eadg3665 (2024).
19. Geng, C. *et al.* Noradrenergic inputs from the locus coeruleus to anterior piriform cortex and the olfactory bulb modulate olfactory outputs. *Nat. Commun.* **16**, 260 (2025).
20. Feng, J. *et al.* A Genetically Encoded Fluorescent Sensor for Rapid and Specific In Vivo Detection of Norepinephrine. *Neuron* **102**, 745-761.e8 (2019).
21. Pirhayati, D. *et al.* Dense and Persistent Odor Representations in the Olfactory Bulb of Awake Mice. *J. Neurosci.* **44**, e0116242024 (2024).
22. Kato, H. K., Chu, M. W., Isaacson, J. S. & Komiyama, T. Dynamic Sensory Representations in the Olfactory Bulb: Modulation by Wakefulness and Experience. *Neuron* **76**, 962–975 (2012).
23. Burton, S. D. *et al.* Mapping odorant sensitivities reveals a sparse but structured representation of olfactory chemical space by sensory input to the mouse olfactory bulb. *eLife* **11**, e80470 (2022).
24. Shani-Narkiss, H., Beniaguev, D., Segev, I. & Mizrahi, A. Stability and flexibility of odor representations in the mouse olfactory bulb. *Front. Neural Circuits* **17**, 1157259 (2023).
25. Liu, G. *et al.* Target specific functions of EPL interneurons in olfactory circuits. *Nat. Commun.* **10**, 3369 (2019).
26. Kudryavitskaya, E., Marom, E., Shani-Narkiss, H., Pash, D. & Mizrahi, A. Flexible categorization in the mouse olfactory bulb. *Curr. Biol.* **31**, 1616-1631.e4 (2021).
27. Wachowiak, M. *et al.* Optical Dissection of Odor Information Processing In Vivo Using GCaMPs Expressed in Specified Cell Types of the Olfactory Bulb. *J. Neurosci* **33**, 5285–5300 (2013).

28. Imamura, F., Ito, A. & LaFever, B. J. Subpopulations of Projection Neurons in the Olfactory Bulb. *Front Neural Circuit* **14**, 561822 (2020).
29. Nagayama, S., Homma, R. & Imamura, F. Neuronal organization of olfactory bulb circuits. *Front Neural Circuit* **8**, 98 (2014).
30. Devore, S. & Linster, C. Noradrenergic and cholinergic modulation of olfactory bulb sensory processing. *Front Behav Neurosci* **6**, 52 (2012).
31. Nai, Q., Dong, H. W., Linster, C. & Ennis, M. Activation of $\alpha 1$ and $\alpha 2$ noradrenergic receptors exert opposing effects on excitability of main olfactory bulb granule cells. *Neuroscience* **169**, 882–892 (2010).
32. Paeger, L. *et al.* Antagonistic modulation of NPY/AgRP and POMC neurons in the arcuate nucleus by noradrenalin. *eLife* **6**, 166 (2017).
33. Yin, X. *et al.* Delayed motor learning in a 16p11.2 deletion mouse model of autism is rescued by locus coeruleus activation. *Nat. Neurosci.* **24**, 646–657 (2021).
34. Gomez, J. L. *et al.* Chemogenetics revealed: DREADD occupancy and activation via converted clozapine. *Science* **357**, 503–507 (2017).
35. Zerbi, V. *et al.* Rapid Reconfiguration of the Functional Connectome after Chemogenetic Locus Coeruleus Activation. *Neuron* (2019) doi:10.1016/j.neuron.2019.05.034.

REVIEWERS' COMMENT

Reviewer #1 (Remarks to the Author):

Authors addressed this reviewer's concerns on data from mouse models, sufficiently.

We thank the reviewer for the note that we have satisfactorily addressed all concerns on the data from the mouse model used in our study.

As for data from human study, I still have concerns since authors did not provide additional evidence. Instead, authors claimed that they did not have samples to perform required experiments or enough data points to reach statistical significance.

We thank the reviewer for pointing out the limitations of our study. In the following we answer to the comment separately, referencing the independent concerns raised.

We have furthermore addressed these critical points on the human study data more thoroughly now in the discussion.

Instead, authors claimed that they did not have samples to perform required experiments....

Referring to the reviewer's "**Specific comment 6)**", we do agree that these data lack a quantitative analysis of the pathology and furthermore we understand, that this might not be entirely satisfying. However, and I hope to address the correct concerns here, we did not feel comfortable quantifying the pathology for the following two reasons:

1) We are uncertain, which anatomical part of the LC is contained in the sections specifically. During the AD continuum, there is important evidence that LC degeneration is anatomically complex and neurons may not degenerate uniformly¹⁻³. Recent studies even indicate a hemisphere-dependent difference in LC integrity during disease progression⁴. This aligns with the emerging view, that the LC is a more heterogenous structure than historically suspected⁵. Heterogeneity in the LC is derived from complex afferent and efferent projection patterns with segregated axonal projection pathways or collateral patterns, diverse somato-dendritic inputs and the differential expression of co-transmitters and neuropeptides^{6,7}. Comparing cell numbers in our tissue between subjects is impeded foremost by the lack of anatomical precision and may thus not lead to conclusive and robust results with our available tissue. To present solid data we would need to evaluate the entire LC region of these subjects, which is impossible, since we do not have the entire LC region archived at the Munich Bio Bank. Larger number of subjects to assume equal variability caused by anatomical difference would be an alternative. Our pAD cases are from donors with neuropathologically defined Braak stages 0-I. However, patients de cease typically at later stages and consequently post-mortem tissue from Braak 0-I is rare. In support of our data and to address the reviewer's concerns, we would like to kindly refer to a sophisticated, recent neuropathological study from Beardmore and colleagues, in which the LC has been

studied more systemically along “the course of AD”⁸. Here, neuromelanin⁺ LC cell loss could not be detected at early Braak stages but significantly declined at Braak III-IV.

2) We have convincingly shown an overall lower NET⁺ axon density in the OB from AD patients at different stages. In our disease mouse model, axon loss occurs while LC neurons are preserved, which is in line with a recent study evaluating the same mouse model at an advanced age of 24 months⁹. In our study, reduced LC axons density alone is sufficient to contribute to our observed olfactory phenotype in mice. In humans the exact sequence of LC-NA system degeneration is less clear. The small size of the LC and the limited resolution of current multimodal imaging data in humans complicates the detailed analysis of a spatiotemporal relationship between axonal and somatic LC degeneration in humans. That is, alterations in cortical NA are often detected in correlation with changes in the LC (volume/integrity) in live human patients. Whether LC axon loss in forebrain regions, however, precedes neurodegeneration in the LC itself has not been answered clearly yet.

Collectively, we fully agree to the reviewer’s comment that LC evaluation in subjects would be critical and interesting in terms of increasing the translational and mechanistic value of the study. To account for the reviewer’s concern, we have now counted the cell numbers and present these data here to the reviewer as count of neuromelanin⁺ neurons in our post-mortem tissue (two-way ANOVA, Tukey’s multiple comparison). While these data show the trend to recapitulate the data mentioned above from Beardmore and colleagues, it also illustrates that the tissue available is anatomically not

ideal to assess cell loss in the LC. Therefore, we solely used the tissue to address the co-pathology in the LC of these brain donors. Furthermore, to address the reviewer’s concerns, we have added that to the discussion of our manuscript addressing the limitations of our current study and to foster future research in the field.

.... or enough data points to reach statistical significance.

We thank the reviewer for the time to evaluate our revised manuscript. With respect to “**Specific comment 7**” from reviewer #1, which we think the remaining concerns relate to, we would like to state, that this is correct. Our human study cohort, which performed olfactory testing in conjunction with TSPO-PET imaging is relatively small compared to the papers cited in the manuscript¹⁰⁻²². We would like to clarify, that our data have been acquired independently in this cohort, prior to the data from our mouse model presented here in our study. We have addressed these concerns in the discussion and we present here the correlation of TSPO-PET and Olfactory Score to the reviewer (one-way *Pearsson* correlation). In any case, we can only speculate if TSPO-PET detected increase in OB

microglia might contribute to any olfactory deficits. Even with a strong and significant correlation with olfactory test scores, we would not be able to relate any causation between these two variables. In addition, since these data are from live patients, correlation with NET+ fiber density in the OB from post-mortem tissue cannot be performed. As mentioned above however, we fully agree to the reviewer's concerns, and we would like to mention, that we and our colleagues are planning to continue our research

in the field aiming to recruit in future studies human patients that will undergo fMRI imaging in conjunction with olfactory testing. In a planned collaborative effort with our colleagues from LMU University Hospital, we seek to develop a pipeline to extract imaging data to measure LC integrity by BOLD and DTI structural connectivity. Upon the successful development of an appropriate code, performing olfactory testing in combination with structural and functional connectivity between the LC and olfactory regions in humans including the olfactory bulb, anterior olfactory nucleus, piriform cortex and secondary olfactory information processing hubs, such as the amygdala or the hippocampus future experiments addressing the critical points mentioned here could become feasible. At the given time, however, we can only present a more detailed discussion in the manuscript referring to and proposing exactly these experiments. We have added this in conjunction with the addressed limitations of our study mentioned in the response above. Collectively, we hope to have at least taken the opportunity to respectfully address the reviewer's concerns in the Discussion of the study.

Reviewer #2 (Remarks to the Author):

All my comments have been addressed satisfactorily. I commend the authors for their work

David Owen

Reviewer #3 (Remarks to the Author):

The authors have fully addressed my concerns. The study is innovative, translational, and ready for publication; I recommend acceptance.

Reviewer #4 (Remarks to the Author):

The authors have strengthened their conclusions in the revised manuscript. In my view, it is now ready for publication.

Dear David Owen, dear Marc Aurel Busche, dear Reviewer #4:

We would like to sincerely thank you for the effort and time invested in evaluating our manuscript. We feel your comments have helped to improve the study significantly and we are excited that you recommend our manuscript to be accepted for publication in *Nature Communications*.

Sincerely and with best regards,

References

1. Jacobs, H. I. L., Becker, A., Riphagen, J. M., Thibault, E. G., Farrell, M. E., Properzi, M. J., Rentz, D. M., Sperling, R. A. & Johnson, K. A. Locus coeruleus integrity exhibits distinct anatomic vulnerabilities to regional tau and amyloid accumulation: parallel and intersecting mechanisms? *Alzheimer's Dement.* **19**, (2023).
2. Riphagen, J. M., Hooren, R. W. E. van, Pagen, L. H. G., Poser, B. A. & Jacobs, H. I. L. Rostro-caudal locus coeruleus integrity differences vary with age and sex using ultra-high field imaging. *Alzheimer's Dement.* **16**, (2020).
3. Kooops, E. A., Dutta, J., Becker, A., Hanseeuw, B. J., Sperling, R. A., Johnson, K. A. & Jacobs, H. I. L. Rostral locus coeruleus metabolism relates to Alzheimer's disease pathology and cognition in amyloid-positive symptomatic individuals. *Alzheimer's Dement.* **20**, e093205 (2025).
4. Beckers, E., Riphagen, J. M., Egroo, M. V., Bennett, D. A. & Jacobs, H. I. L. Sparse Asymmetry in Locus Coeruleus Pathology in Alzheimer's Disease. *J. Alzheimer's Dis.* **99**, 105–111 (2024).
5. Ma, H., Zhang, H., Zuo, Z. & Liu, Y. Heterogeneous organization of Locus coeruleus: An intrinsic mechanism for functional complexity. *Physiol. Behav.* **268**, 114231 (2023).
6. Schwarz, L. A. & Luo, L. Organization of the locus coeruleus-norepinephrine system. *Current biology : CB* **25**, R1051-6 (2015).
7. Schwarz, L. A., Miyamichi, K., Gao, X. J., Beier, K. T., Weissbourd, B., DeLoach, K. E., Ren, J., Ibanes, S., Malenka, R. C., Kremer, E. J. & Luo, L. Viral-

genetic tracing of the input-output organization of a central noradrenaline circuit. *Nature* **524**, 88–92 (2015).

8. Beardmore, R., Durkin, M., Zayee-Mellick, F., Lau, L. C., Nicoll, J. A. R., Holmes, C. & Boche, D. Changes in the locus coeruleus during the course of Alzheimer's disease and their relationship to cortical pathology. *Neuropathol. Appl. Neurobiol.* **50**, e12965 (2024).

9. Sakakibara, Y., Hirota, Y., Ibaraki, K., Takei, K., Chikamatsu, S., Tsubokawa, Y., Saito, T., Saido, T. C., Sekiya, M. & Iijima, K. M. Widespread Reduced Density of Noradrenergic Locus Coeruleus Axons in the App Knock-In Mouse Model of Amyloid- β Amyloidosis. *J Alzheimer's Dis* 1–18 (2021). doi:10.3233/jad-210385

10. Woodward, M. R., Amrutkar, C. V., Shah, H. C., Benedict, R. H. B., Rajakrishnan, S., Doody, R. S., Yan, L., Szigeti, K., Consortium, T. A. R. and C., Dang, M. M., Pavlik, V., Chan, W., Massman, P., Darby, E., Evans, T., Khaleeq, A., Wu, C.-K., Lambert, M., Perez, V., Hernandez, M., Fairchild, T., Knebl, J., O'Bryant, S. E., Hall, J. R., Johnson, L., Barber, R. C., Mains, D., Alvarez, L., McCallum, R., Adams, P., Cullum, M., Rosenberg, R., Williams, B., Quiceno, M., Reisch, J., Huebinger, R., Martinez, N., Smith, J., Royall, D., Palmer, R., Polk, M., Sohrabji, F., Balsis, S., Miranda, R., Waring, S. C., Wilhelmsen, K. C., Tilson, J. L. & Chasse, S. Validation of olfactory deficit as a biomarker of Alzheimer disease. *Neurol.: Clin. Pr.* **7**, 5–14 (2017).

11. Guo, J., Dove, A., Wang, J., Laukka, E. J., Ekström, I., Dunk, M. M., Bennett, D. A. & Xu, W. Trajectories of olfactory identification preceding incident mild cognitive impairment and dementia: a longitudinal study. *eBioMedicine* **98**, 104862 (2023).

12. Pacyna, R. R., Han, S. D., Wroblewski, K. E., McClintock, M. K. & Pinto, J. M. Rapid olfactory decline during aging predicts dementia and GMV loss in AD brain regions. *Alzheimer's Dement.* **19**, 1479–1490 (2023).

13. Audronyte, E., Pakulaite-Kazliene, G., Sutnikiene, V. & Kaubrys, G. Properties of odor identification testing in screening for early-stage Alzheimer's disease. *Sci. Rep.* **13**, 6075 (2023).

14. Wheeler, P. L. & Murphy, C. Olfactory Measures as Predictors of Conversion to Mild Cognitive Impairment and Alzheimer's Disease. *Brain Sci.* **11**, 1391 (2021).

15. Liu, D., Lu, J., Wei, L., Yao, M., Yang, H., Lv, P., Wang, H., Zhu, Y., Zhu, Z., Zhang, X., Chen, J., Yang, Q. X. & Zhang, B. Olfactory deficit: a potential functional marker across the Alzheimer's disease continuum. *Front. Neurosci.* **18**, 1309482 (2024).

16. Igeta, Y., Hemmi, I., Yuyama, K. & Ouchi, Y. Odor identification score as an alternative method for early identification of amyloidogenesis in Alzheimer's disease. *Sci. Rep.* **14**, 4658 (2024).

17. Audronyte, E., Pakulaite-Kazliene, G., Sutnickiene, V. & Kaubrys, G. Odor Discrimination as a Marker of Early Alzheimer's Disease. *J. Alzheimer's Dis.* **94**, 1169–1178 (2023).
18. Dintica, C. S., Marseglia, A., Rizzuto, D., Wang, R., Seubert, J., Arfanakis, K., Bennett, D. A. & Xu, W. Impaired olfaction is associated with cognitive decline and neurodegeneration in the brain. *Neurology* **92**, 10.1212/WNL.0000000000006919 (2019).
19. Liu, S., Jiang, Z., Zhao, J., Li, Z., Li, R., Qiu, Y. & Peng, H. Disparity of smell tests in Alzheimer's disease and other neurodegenerative disorders: a systematic review and meta-analysis. *Front. Aging Neurosci.* **15**, 1249512 (2023).
20. Delgado-Lima, A. H., Bouhaben, J., Martínez-Zujeros, S., Pallardo-Rodil, B., Gómez-Pavón, J. & Delgado-Losada, M. L. Could olfactory identification be a prognostic factor in detecting cognitive impairment risk in the elderly? *GeroScience* **45**, 2011–2025 (2023).
21. Tian, Q., Bilgel, M., Moghekar, A. R., Ferrucci, L. & Resnick, S. M. Olfaction, Cognitive Impairment, and PET Biomarkers in Community-Dwelling Older Adults. *J. Alzheimer's Dis.* **86**, 1275–1285 (2022).
22. Shrestha, S., Zhu, X., Griswold, M. E., Palta, P., Sullivan, K. J., Chen, H., Schneider, A. L. C., Moghekar, A., Grove, M. L., Thyagarajan, B., Pike, J. R., Gottesman, R. F., Windham, B. G., Mosley, T. H., Deal, J. A. & Kamath, V. Olfaction and Plasma Biomarkers of Alzheimer Disease and Neurodegeneration in the Atherosclerosis Risk in Communities Study. *Neurology* **104**, e213706 (2025).